# MOTS-c promotes phosphorodiamidate morpholino oligomer uptake and efficacy in dystrophic mice

Ning Ran[1],[†], Caorui Lin[1],[†], Ling Leng[1], Gang Han[2], Mengyuan Geng[1], Yingjie Wu[1], Scott Bittner[3], Hong M Moulton[3] & HaiFang Yin[1,2,4,*] (ID)

## Abstract

**Antisense oligonucleotide (AO)-mediated exon-skipping therapies show promise in Duchenne muscular dystrophy (DMD), a devastating muscular disease caused by frame-disrupting mutations in the *DMD* gene. However, insufficient systemic delivery remains a hurdle to clinical deployment. Here, we demonstrate that MOTS-c, a mitochondria-derived bioactive peptide, with an intrinsic muscle-targeting property, augmented glycolytic flux and energy production capacity of dystrophic muscles *in vitro* and *in vivo*, resulting in enhanced phosphorodiamidate morpholino oligomer (PMO) uptake and activity in *mdx* mice. Long-term repeated administration of MOTS-c (500 μg) and PMO at the dose of 12.5 mg/kg/week for 3 weeks followed by 12.5 mg/kg/month for 3 months (PMO-M) induced therapeutic levels of dystrophin expression in peripheral muscles, with up to 25-fold increase in diaphragm of *mdx* mice over PMO alone. PMO-M improved muscle function and pathologies in *mdx* mice without detectable toxicity. Our results demonstrate that MOTS-c enables enhanced PMO uptake and activity in dystrophic muscles by providing energy and may have therapeutic implications for exon-skipping therapeutics in DMD and other energy-deficient disorders.**

**Keywords** duchenne muscular dystrophy; energy; exon-skipping; MOTS-c; PMO

**Subject Category** Musculoskeletal System

## Introduction

Duchenne muscular dystrophy (DMD) is a debilitating muscular disorder caused by frame-shifting mutations in the *DMD* gene. These mutations result in premature termination of protein translation and consequently the absence or truncation of functional dystrophin (Verhaart & Aartsma-Rus, 2019). Currently, there is no cure available in the clinic. Although antisense oligonucleotide (AO)-mediated exon-skipping therapies show promise in DMD, with AO drugs including eteplirsen, golodirsen, and viltolarsen approved by the US FDA and in Japan (Syed, 2016; Roshmi & Yokota, 2019; Frank *et al*, 2020), the limited systemic efficacy of these drugs due to insufficient systemic delivery (Godfrey *et al*, 2017) can be improved. Therefore, approaches to enhance AO delivery are vital for clinical deployment of AOs in DMD.

Different strategies have been studied to enhance AO delivery in DMD (Juliano, 2016). For instance, cell-penetrating, muscle-targeting, and chimeric peptides were tested to facilitate the delivery of phosphorodiamidate morpholino oligomer (PMO) to muscle via covalent conjugation (Moulton *et al*, 2007; Yin *et al*, 2009; Gao *et al*, 2014); however, the safety profiles for these peptides remain to be established. Lipid-, polymer- and exosome-based nanoparticles have also been attempted (Juliano, 2016; Gao *et al*, 2018), though these are still at early developmental stages. In recent years, small molecules, particularly compounds used in the clinic or dietary supplements, have attracted much attention, such as dantrolene (Kendall *et al*, 2012) and hexose (glucose: fructose-GF; Han *et al*, 2016). The former enhanced PMO activity by augmenting its exon-skipping frequency, whereas the latter promoted PMO uptake via energy replenishment in energy-deficient dystrophic muscles. Although these molecules show potential in potentiating PMO activity, one needs to be cautious about the side effects associated with dantrolene (Wedel *et al*, 1995); and insulin resistance associated with a subpopulation of DMD patients (Rodriguez-Cruz *et al*, 2015) requires careful titration of GF (glucose: fructose-1:1). Therefore, other delivery strategies need to be pursued.

MOTS-c is a 16-amino acid mitochondria-derived bioactive peptide and plays an important role in metabolic homeostasis by regulating insulin resistance, obesity and inhibiting inflammation (Lee *et al*, 2015; Lee *et al*, 2016; Yan *et al*, 2019). Importantly, MOTS-c can enhance glucose utilization and glycolytic flux, and thus contributes to energy production (Lee *et al*, 2015). Here, we

1 Tianjin Key Laboratory of Cellular Homeostasis and Human Diseases & The Province and Ministry Co-sponsored Collaborative Innovation Center for Medical Epigenetics &, Department of Cell Biology, Tianjin Medical University, Tianjin, China
2 School of Medical Laboratory, Tianjin Medical University, Tianjin, China
3 Biomedical Sciences, College of Veterinary Medicine, Oregon State University, Corvallis, OR, USA
4 Department of Neurology, Tianjin Medical University General Hospital, Tianjin, China
*Corresponding author. Tel: +86 22 83336537; E-mail: haifangyin@tmu.edu.cn
†These authors contributed equally to this work

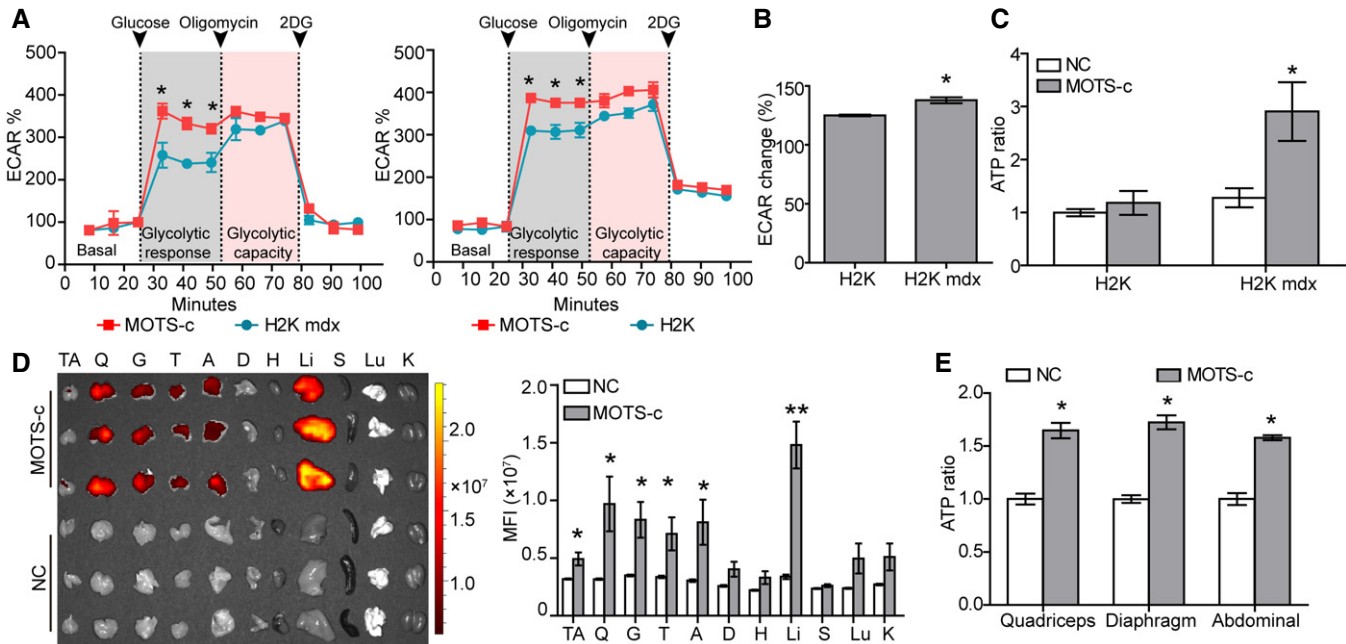

**Figure 1. Evaluation on energy production and muscle-targeting capacity of MOTS-c in dystrophic muscles *in vitro* and *in vivo*.**

A   Measurement of maximum glycolytic capacity of MOTS-c in dystrophic muscle cells after oligomycin and 2-DG treatments ($n = 6$; *$P < 0.05$).
B   Quantification of ECAR changes in relative to basal ECAR in dystrophic muscle cells treated with MOTS-c ($n = 6$; *$P < 0.05$).
C   ATP assay for dystrophic muscle cells in the presence of MOTS-c ($n = 6$; *$P < 0.05$). NC refers to untreated H2K or H2K mdx cells.
D   Tissue distribution and quantitative analysis of Rhodamine B-labeled MOTS-c in *mdx* mice after single intravenous injection (500 µg) ($n = 6$; *$P < 0.05$, *$P < 0.001$). Tissues were harvested and examined 2 h after injection. This experiment was repeated twice. NC refers to untreated *mdx* mice. TA—tibialis anterior, Q—quadriceps, G—gastrocnemius, T—triceps, A—abdominal muscle, D—diaphragm, H—heart, Li—liver, S—spleen, Lu—lung, and K—kidney.
E   Measurement of ATP levels in peripheral muscles of *mdx* mice after single intravenous injection of MOTS-c (500 µg) ($n = 6$; *$P < 0.05$).

Data information: Data were presented as mean ± sem; two-tailed *t*-test was used for statistical analysis. Exact *P* values are specified in Appendix Table S1.

demonstrate that MOTS-c significantly augmented glycolytic rate and adenosine triphosphate (ATP) production in energy-deficient dystrophic muscles. Co-administration of MOTS-c with PMO resulted in enhanced PMO uptake and activity in dystrophic muscles *in vitro* and *in vivo*. Moreover, repeated administration of MOTS-c

with extremely low doses of PMO (12.5 mg/kg) elicited therapeutic levels of dystrophin restoration and functional improvements in *mdx* mice without any detectable adverse effect, though more comprehensive toxicological studies are required prior to clinical deployment.

**Figure 2. Effect of MOTS-c on PMO uptake in dystrophic muscles *in vitro* and *in vivo*.**

A   Measurement of PMO uptake in dystrophic muscle cells (H2K *mdx* cells) in the presence of MOTS-c (scale bar: 10 µm) ($n = 6$; **$P < 0.001$). PMO-M refers to PMO and MOTS-c (the same is for the rest unless otherwise specified).
B   Tissue distribution of FITC-labeled PMO (50 mg/kg) mixed with MOTS-c (500 µg) in *mdx* mice 48 h after single intravenous injection. NC represents untreated *mdx* control. The abbreviation is the same as Fig 1D. Color scale represents the fluorescence intensity.
C   Quantification of fluorescence intensity in body-wide tissues of *mdx* mice treated with PMO-M or PMO alone ($n = 3$; *$P < 0.05$). The comparison was conducted between PMO-M and PMO.
D, E   Immunohistochemistry (D) and quantitative analysis (E) of dystrophin-positive fibers in *mdx* TA muscles treated with single intramuscular injection of 0.5 µg PMO mixed with MOTS-c (20 µg) (PMO-M) or PMO alone (scale bar: 100 µm) ($n = 3$; **$P < 0.001$). C57 means wild-type control *C57BL/6*.
F   Representative Western blot and quantitative analysis for dystrophin expression in TA muscles from *mdx* mice treated with single intramuscular injection of PMO-M or PMO alone ($n = 3$; *$P < 0.05$). 0.5 µg and 2.5 µg total protein from *C57BL/6* and 50 µg from muscle samples of untreated and treated *mdx* mice were loaded. α-actinin was used as the loading control. TA muscles from *C57BL/6* were used as normal controls (the same is for all Western blots unless otherwise specified).
G   Systemic evaluation of MOTS-c (500 µg) and PMO at the dose of 50 mg/kg/week for 3 weeks in adult *mdx* mice intravenously. Tissues were examined two weeks after last injection. Immunohistochemistry for dystrophin-positive fibers in body-wide muscles of treated *mdx* mice (scale bar: 100 µm). C57 means wild-type control *C57BL/6*.
H, I   Western blot (H) and quantitative analysis (I) for dystrophin expression in body-wide muscles from treated *mdx* mice ($n = 3$; *$P < 0.05$). 0.5 µg, 2.5 µg, 5 µg, and 10 µg total protein from *C57BL/6* and 20 µg of muscle samples from untreated and treated *mdx* mice were loaded. α-actinin was used as the loading control.

Data information: Data were presented as mean ± sem; two-tailed *t*-test was used for statistical analysis. Exact *P* values are specified in Appendix Table S1.

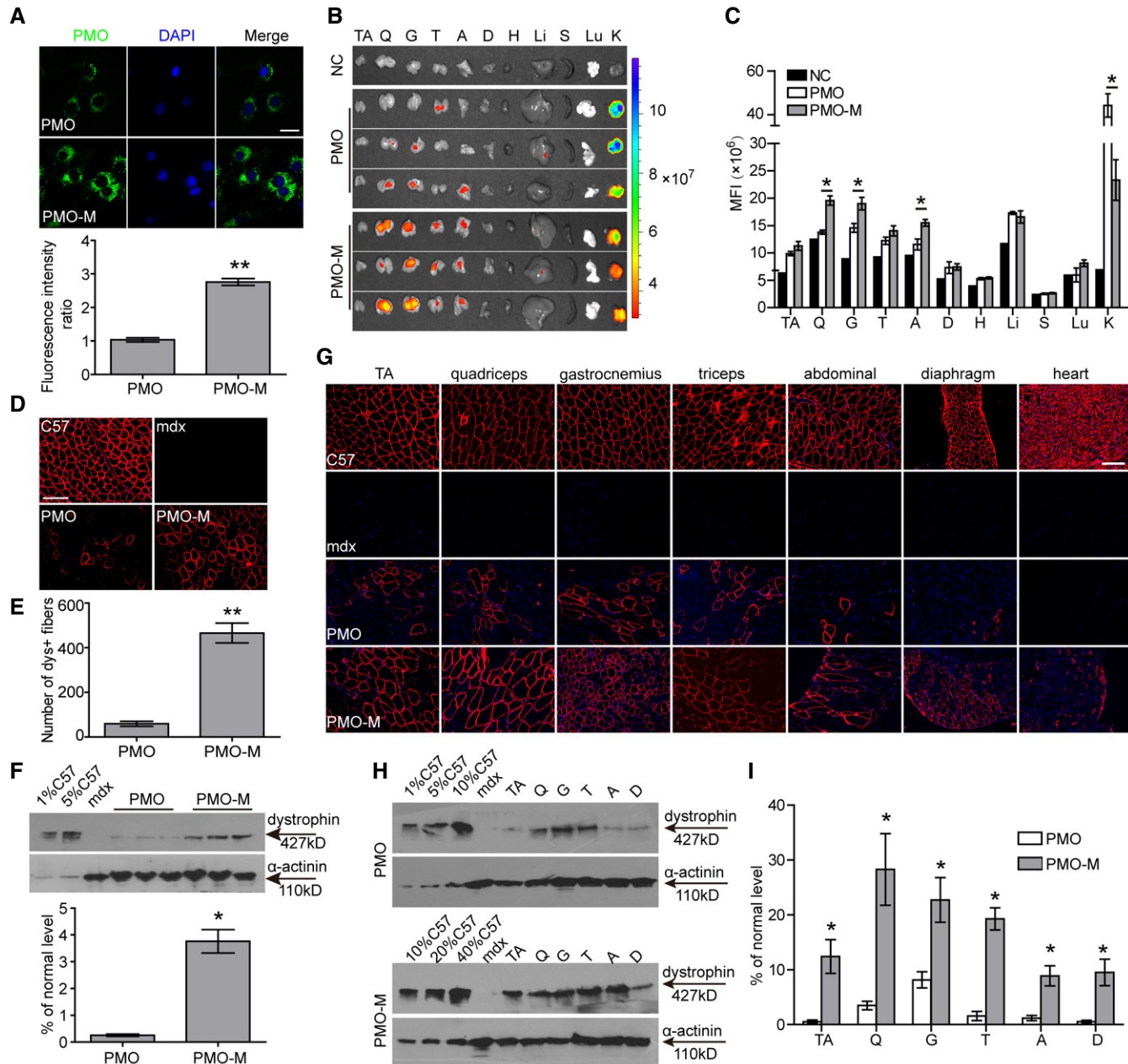

**Figure 2.**

## Results

### MOTS-c triggers increased energy production in dystrophic muscles

As MOTS-c is known to stimulate glycolysis and increase ATP levels in HEK293 cells (Lee *et al*, 2015), to determine whether a similar effect can be observed in energy-deficient dystrophic muscle cells (Han *et al*, 2016), we examined the extracellular acidification rate (ECAR), indicative of glycolytic flux (TeSlaa & Teitell, 2014), in the supernatant of H2K *mdx* cells. A significantly increased ECAR was observed in H2K *mdx* cells with the addition of MOTS-c compared to cells alone in the presence of glucose (Fig 1A) when the

glycolysis stress was induced with oligomycin, an ATP synthase inhibitor (Shchepina *et al*, 2002), and 2-deoxyglucose (2-DG), an inhibitor for glycolysis (Zhong *et al*, 2009), sequentially. This result demonstrates that the increase in ECAR upon addition of glucose is due to anaerobic glycolysis rather than other sources. Notably, a greater glycolytic flux was found in H2K *mdx* cells than normal H2K cells in the presence of MOTS-c as evidenced by significantly increased ECAR (Fig 1B). Importantly, a significantly greater level of ATP was detected in H2K *mdx* cells than normal H2K cells in the presence of MOTS-c (Fig 1C), suggesting that MOTS-c stimulates increased glycolysis and ATP production in dystrophic muscle cells. Strikingly, a substantial amount of MOTS-c was found in peripheral muscles when Rhodamine B-labeled MOTS-c was administered into

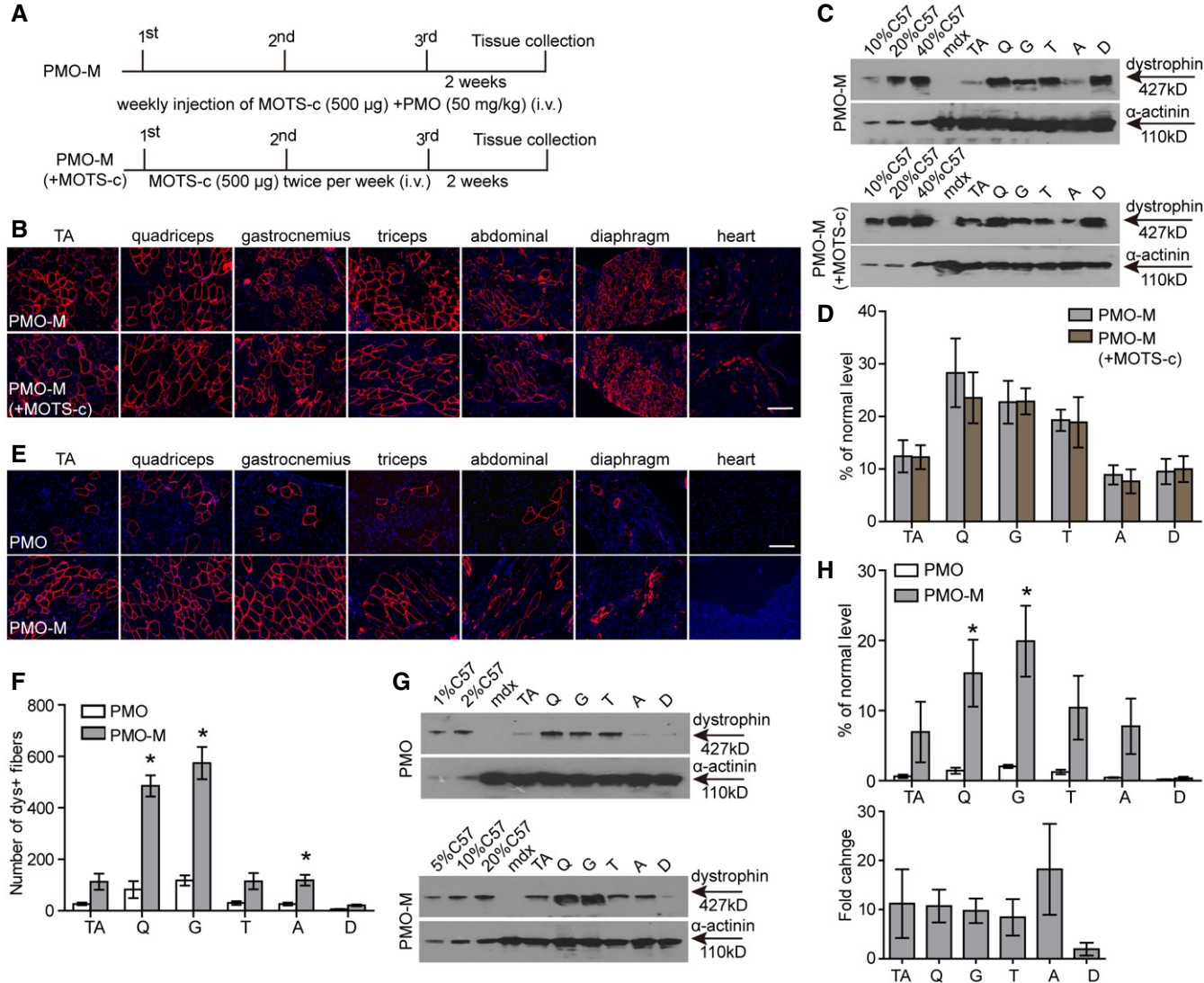

**Figure 3. Investigation on the dose-dependent effect of MOTS-c on PMO activity in adult *mdx* mice.**

A Diagram of dosing regimen for the dose-dependent effect of MOTS-c on PMO activity in *mdx* mice. i.v. refers to intravenous injection. PMO-M (+MOTS-c) means PMO-M supplemented with additional MOTS-c.

B Immunohistochemistry for dystrophin expression in body-wide muscles from *mdx* mice treated with PMO-M or PMO-M (+MOTS-c) (scale bar: 100 μm). TA refers to tibialis anterior.

C, D Western blot (C) and quantitative analysis (D) for dystrophin expression in body-wide muscles from *mdx* mice treated with PMO-M or PMO-M (+MOTS-c) (*n* = 3). 5 μg, 10 μg, and 20 μg total protein from *C57BL/6* and 50 μg of muscle samples from untreated and treated *mdx* mice were loaded.

E, F Immunohistochemistry (E) and quantitative analysis (F) of dystrophin-positive fibers in body-wide muscles from *mdx* mice treated with PMO at 12.5 mg/kg/week for 3 weeks mixed with MOTS-c (500 μg) (PMO-M) intravenously (scale bar: 100 μm) (*n* = 3; *$P$ < 0.05). PMO represents PMO in saline.

G, H Western blot (G) and quantitative analysis (H) of dystrophin expression in body-wide muscles from *mdx* mice treated with PMO at 12.5 mg/kg/week for 3 weeks mixed with MOTS-c (500 μg) intravenously (*n* = 3; *$P$ < 0.05). 0.5 μg, 1 μg, 2.5 μg, 5 μg, and 10 μg total protein from *C57BL/6* and 50 μg of muscle samples from untreated and treated *mdx* mice were loaded. α-actinin was used as the loading control. Fold change refers to PMO-M relative to PMO alone.

Data information: Data were presented as mea*n* ± sem; two-tailed *t*-test was used for statistical analysis. Exact *P* values are specified in Appendix Table S1.

*mdx* mice intravenously at single dose of 500 μg (Fig 1D), indicating that MOTS-c primarily targets to skeletal muscles. Consistent with the previous report (Lee *et al*, 2015), MOTS-c also accumulated in the liver (Fig 1D). Corroborating with *in vitro* data, significantly increased levels of ATP were observed in body-wide muscles (Fig 1E). These results showed that MOTS-c triggers greater glycolysis and energy production in dystrophic muscle cells *in vitro* and *in vivo*.

**MOTS-c promotes PMO uptake in dystrophic muscles *in vitro* and *in vivo***

Since MOTS-c enabled greater levels of ATP production in dystrophic muscle cells and increased ATP availability was shown to enhance PMO uptake in energy-deficient dystrophic muscles (Han *et al*, 2016), we hypothesized that MOTS-c might be able to facilitate PMO uptake in dystrophic muscle cells. As expected, the

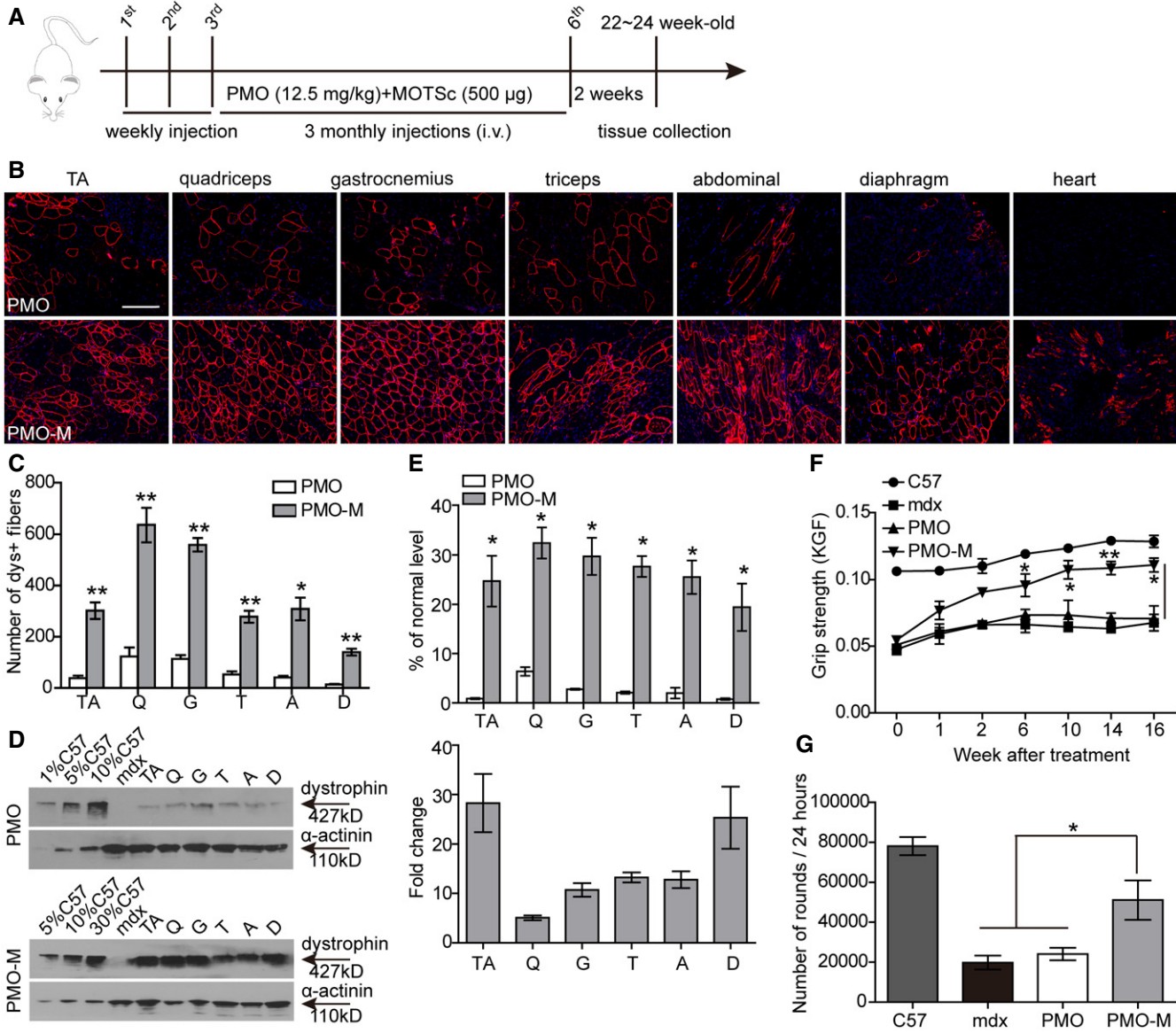

**Figure 4. Long-term repeated administration of PMO-M in adult *mdx* mice.**

PMO-M was administered intravenously into adult *mdx* mice at the PMO dose of 12.5 mg/kg/week for 3 weeks followed by 12.5 mg/kg/month for 3 months, and tissues were harvested two weeks after last injection.

A    Diagram of dosing regimen for the long-term study of PMO-M in *mdx* mice. i.v. refers to intravenous injection.

B, C    Immunohistochemistry (B) and quantitative analysis (C) of dystrophin-positive fibers in body-wide muscles from *mdx* mice treated with repeated doses of PMO-M (*n* = 4) or PMO alone (*n* = 3) (scale bar: 100 μm) (*P < 0.05, **P < 0.001; two-tailed *t*-test).

D    Western blot for dystrophin expression in body-wide muscles from *mdx* mice treated with PMO-M or PMO alone. 0.5 μg, 2.5 μg, 5 μg, and 15 μg total protein from *C57BL/6* and 50 μg of muscle samples from untreated and treated *mdx* mice were loaded.

E    Quantitative analysis of dystrophin expression in body-wide muscles from *mdx* mice treated with PMO-M (*n* = 4) or PMO alone (*n* = 3) (*P < 0.05; two-tailed *t*-test).

F    Muscle function was assessed to determine the physical improvement with grip strength test for *mdx* mice treated with PMO-M (*n* = 4) or PMO alone (*n* = 3), untreated *mdx* controls (*n* = 3), and *C57BL/6* (*n* = 3) (*P < 0.05, **P < 0.001, one-way ANOVA post hoc Student–Newman–Keuls test).

G    Measurement of muscle endurance with the running wheel test for wild-type *C57BL/6* (C57) (*n* = 3), untreated *mdx* controls (mdx) (*n* = 3), and *mdx* mice treated with PMO-M (*n* = 4) or PMO alone (*n* = 3) (*P < 0.05, one-way ANOVA post hoc Student–Newman–Keuls test). Muscle functional tests were conducted two weeks after last injection.

Data information: Data were presented as mea*n* ± sem. Exact *P* values are specified in Appendix Table S1.

fluorescence was significantly increased in H2K *mdx* cells when fluorescein isothiocyanate (FITC)-labeled PMO was added together with MOTS-c compared to cells treated with PMO alone, with PMO primarily located in the cytoplasm (Fig 2A). To further investigate if MOTS-c can enhance PMO uptake in dystrophic muscle cells *in vivo*, we intravenously injected MOTS-c (500 μg) with FITC-

labeled PMO at single dose of 50 mg/kg, a dose adopted from previous studies (Han *et al*, 2016; Zhai *et al*, 2017), into adult *mdx* mice. Significantly increased fluorescence was found in peripheral muscles of *mdx* mice treated with PMO and MOTS-c (PMO-M) compared to PMO alone and untreated *mdx* controls (Fig 2B and C). Consistently, examination of PMO in peripheral muscles of treated *mdx* mice revealed significantly increased amounts of PMO in tibialis anterior (TA) muscles and gastrocnemius of PMO-M-treated *mdx* mice compared to PMO-treated *mdx* mice (Fig EV1), indicating that MOTS-c promotes PMO uptake in dystrophic muscles. Commensurate with enhanced PMO uptake in dystrophic muscles, a substantial number of dystrophin-positive myofibers and significantly greater level of dystrophin expression were detected in TA muscles of *mdx* mice treated with PMO (0.5 μg) and MOTS-c (20 μg) (PMO-M) intramuscularly compared to PMO alone (Fig 2D–F). To exclude the possibility that MOTS-c has a direct impact on PMO uptake, we administered the same amount of PMO (0.5 μg) into TA muscles of *mdx* mice, followed by simultaneous intravenous injection of MOTS-c (500 μg) into the same *mdx* mice. A similar level of dystrophin expression was observed in TA muscles treated with intramuscular PMO combined with intravenous MOTS-c compared to PMO-M (Fig EV2A–D), suggesting that MOTS-c functions by promoting PMO uptake rather than through direct interaction with PMO. To determine the systemic effect of MOTS-c on PMO activity, we intravenously injected 500 μg MOTS-c with PMO (PMO-M) at the dose of 50 mg/kg/week for 3 weeks in *mdx* mice, a dosing regimen used for our previous study (Han *et al*, 2016). Strikingly, a profound enhancement was observed in *mdx* mice treated with PMO-M as revealed by more uniform distribution of dystrophin-positive myofibers in peripheral muscles except for the heart (Fig 2G) and greater levels of dystrophin restoration (Fig 2H and I), compared to PMO alone under identical conditions. Importantly, a correlation was established between the level of dystrophin expression in different muscles (except for abdominal muscles) of *mdx* mice treated with PMO-M intravenously at the PMO dose of 50mg/kg/week for 3 weeks and distribution of PMO (Fig EV2E). These data indicated that MOTS-c enables efficient PMO uptake in dystrophic muscles *in vitro* and *in vivo*.

## MOTS-c enhances PMO activity in *mdx* mice in a saturable manner

To examine whether there was any cumulative effect, we increased the administration frequency of MOTS-c from weekly to twice per week for 3 weeks (Fig 3A). However, there was no difference in dystrophin expression between *mdx* mice treated by PMO-M with or without additional MOTS-c (Fig 3B–D), suggesting that MOTS-c potentiates PMO efficacy in *mdx* mice in a saturable manner. To investigate whether MOTS-c can lower the systemic dose of PMO, we intravenously injected 500 μg MOTS-c and PMO at a dose of 12.5 mg/kg/week for 3 weeks in *mdx* mice, a dose lower than those used in previous studies (Han *et al*, 2016; Lin *et al*, 2020). Surprisingly, significantly increased numbers of dystrophin-positive myofibers and levels of dystrophin expression were found in peripheral muscles of *mdx* mice treated with PMO-M compared to PMO alone under identical conditions (Fig 3E–H), further confirming the potency of MOTS-c. To determine whether the efficacy depending on the fiber type as MOTS-c primarily provides energy via glycolysis

(Lee *et al*, 2015), we examined the expression of dystrophin in type I myosin heavy chain (MHC)-positive slow-twitch soleus, type IIa and IIb MHC-positive fast-twitch extensor digitorum longus (EDL) and mixed type of muscles (triceps; Schiaffino & Reggiani, 2011; Vila *et al*, 2015; Liu *et al*, 2016). The results showed significant increases in the number of dystrophin-positive fibers in different muscles from PMO-M-treated *mdx* mice compared to PMO-treated *mdx* mice irrespective of fiber types, however, the fold change was much greater in type IIb MHC-positive fast-twitch EDL than type I MHC-positive slow-twitch soleus (Fig EV3A and B), confirming that glycolysis is primarily responsible for energy production, resulting in higher PMO uptake in fast-twitch fibers.

As GF was shown to enhance PMO uptake in dystrophic muscles by replenishing energy stores (Han *et al*, 2016), we wondered whether MOTS-c would outperform GF in restoring dystrophin expression. In comparison with PMO in GF (PMO-GF) under identical conditions, PMO-M elicited more effective dystrophin restoration as demonstrated by dystrophin-positive myofibers and levels of dystrophin expression (Appendix Fig S1A–C), indicating that MOTS-c is more potent than GF in augmenting PMO activity. Collectively, these results indicated that MOTS-c enhances PMO activity in a saturable manner and reduces the required systemic dose of PMO in *mdx* mice.

## PMO-M induces long-term therapeutic efficacy and phenotypic rescue in *mdx* mice

Since PMO-M elicited effective dystrophin restoration at a lower dose, we wished to determine its long-term efficacy in *mdx* mice with a dosing regimen similar to previous studies (Han *et al*, 2016; Lin *et al*, 2020), though the dose of PMO was reduced to 12.5 mg/kg (Fig 4A). Surprisingly, widespread expression of dystrophin over multiple tissue sections within each muscle group was detected in hind limb, forelimb, abdominal wall, and diaphragm muscles, but not in the heart of *mdx* mice treated with repeated injections of PMO-M (Fig 4B and C). Consistently, a significant elevation of dystrophin restoration was observed in body-wide peripheral muscles of *mdx* mice treated with PMO-M compared to PMO alone (Fig 4D and E) except for the heart (Appendix Fig S2), with up to 25-fold higher dystrophin expression found in diaphragm than PMO alone (Fig 4E). Concordantly, examination of dystrophin expression in fast-twitch EDL and slow-twitch soleus muscles revealed significantly increased dystrophin expression in both muscle types from PMO-M-treated *mdx* mice compared to PMO-treated *mdx* mice, though to a greater extent in EDL than soleus muscles (Fig EV4A and B), further confirming that MOTS-c primarily promotes energy production via glycolysis and potentiates PMO activity. Importantly, significant improvements in muscle force and endurance were achieved in *mdx* mice treated with PMO-M compared to PMO alone and untreated age-matched *mdx* controls, demonstrated by grip strength (Fig 4F) and running wheel tests (Fig 4G). Overall, our data demonstrated that PMO-M elicits therapeutic levels of dystrophin restoration and muscle improvement in *mdx* mice at low doses.

## PMO-M improves muscle pathologies without detectable toxicity in *mdx* mice

Corroborating with therapeutic levels of dystrophin restoration, levels of serum creatine kinase (CK), which is usually elevated in DMD patients due to leaky muscle membrane (Kim *et al*, 2017),

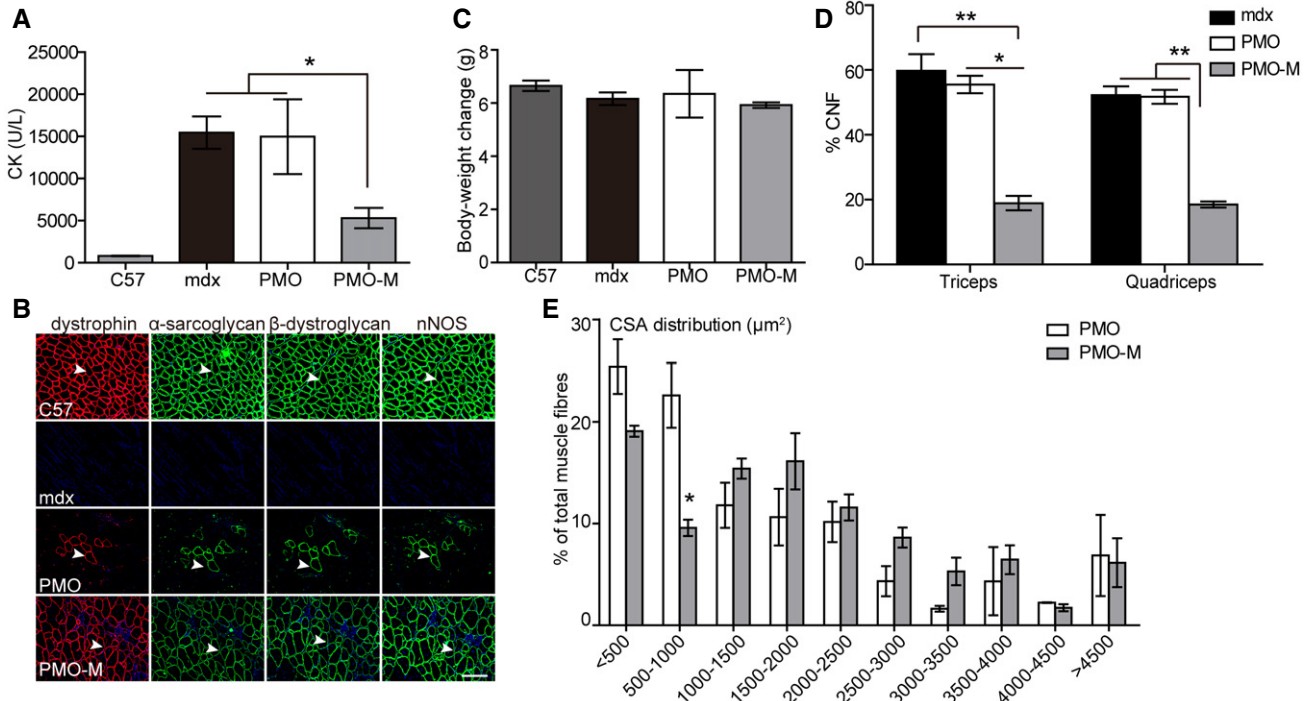

**Figure 5. Morphological assessment of *mdx* mice treated with long-term repeated administration of PMO-M or PMO.**

PMO-M was administered intravenously into adult *mdx* mice at the PMO dose of 12.5 mg/kg/week for 3 weeks followed by 12.5 mg/kg/month for 3 months and tissues / blood were harvested 2 weeks after last injection.

A    Measurement of serum creatine kinase (CK) levels in wild-type *C57BL/6* (C57) (*n* = 3), untreated *mdx* controls (mdx) (*n* = 3), and *mdx* mice treated with PMO-M (*n* = 4) or PMO alone (*n* = 3) (*$P < 0.05$, one-way ANOVA post hoc Student–Newman–Keuls test).

B    Re-localization of DAPC components in treated *mdx* mice to assess dystrophin function and recovery of normal myoarchitecture (scale bar: 100 μm). The arrowheads point to identical myofibers. C57 means wild-type *C57BL/6*; mdx represents untreated *mdx* controls.

C    Measurement of body-weight changes of wild-type *C57BL/6* (C57) (*n* = 3), untreated *mdx* controls (mdx) (*n* = 3), and *mdx* mice treated with PMO-M (*n* = 4) or PMO alone (*n* = 3).

D    Quantitative analysis of centrally nucleated fibers (CNFs) in triceps and quadriceps of wild-type *C57BL/6* (C57) (*n* = 3), untreated *mdx* controls (mdx) (*n* = 3), and *mdx* mice treated with repeated doses of PMO-M (*n* = 4) or PMO (*n* = 3) (*$P < 0.05$, **$P < 0.001$, one-way ANOVA post hoc Student–Newman–Keuls test).

E    Measurement of cross-sectional area (CSA) of muscle fibers from *mdx* mice treated with repeated doses of PMO-M (*n* = 4) or PMO (*n* = 3) (*$P < 0.05$, two-tailed *t*-test).

Data information: Data were presented as mean ± sem. Exact *P* values are specified in Appendix Table S1.

significantly declined in *mdx* mice treated with PMO-M compared to PMO alone and untreated *mdx* controls (Fig 5A). Also, dystrophin-associated protein complex (DAPC), which were mis-localized in the absence of dystrophin in dystrophic muscles (Matsumura *et al*, 1993), were correctly re-localized in sarcolemma of quadriceps from *mdx* mice treated with PMO-M as evidenced by serial staining of α-sarcoglycan, β-dystroglycan, and nNOS (Fig 5B). There were no abnormal behavior and body-weight changes observed in treated *mdx* mice during the period of experiments (Fig 5C), suggesting that MOTS-c does not have a direct impact on body weight at tested doses. Importantly, significantly decreased numbers of centrally nucleated fibers (CNFs), indicative of muscle regeneration (van Putten *et al*, 2012), and less variable cross-sectional area were found in peripheral muscles of PMO-M-treated *mdx* mice compared to PMO-treated *mdx* mice (Fig 5D and E), indicating that PMO-M effectively prevents muscle pathological progression. Levels of circulatory aspartate transaminase (AST) and alanine aminotransferase (ALT), which rise in DMD patients (McMillan *et al*, 2011), were

significantly decreased in *mdx* mice treated with PMO-M, whereas there was no significant change in levels of serum liver enzyme gamma-glutamyl transferase (GGT) (Fujii *et al*, 2020; Fig 6A), creatinine (CREA) and blood urea nitrogen (BUN) (Fig 6B), markers for kidney function (Uchino *et al*, 2012), in *mdx* mice treated with PMO-M compared to PMO alone and untreated *mdx* controls. Consistently, no morphological abnormality was found in kidney and liver (Fig 6C), indicating that PMO-M does not cause any detectable toxicity. Inflammatory conditions were also improved as reflected by significantly decreased numbers of CD3-positive T cells, CD11b-positive inflammatory monocytes, and CD68-positive macrophages which are predominantly present in dystrophic muscles (Geissmann *et al*, 2003; Rosenberg *et al*, 2015), in quadriceps and diaphragm of *mdx* mice treated with PMO-M compared to PMO alone and untreated *mdx* controls (Fig 6D and E). These data strengthen the conclusion that PMO-M elicits therapeutic levels of dystrophin, resulting in muscle pathological mitigation in *mdx* mice without triggering any detectable toxicity.

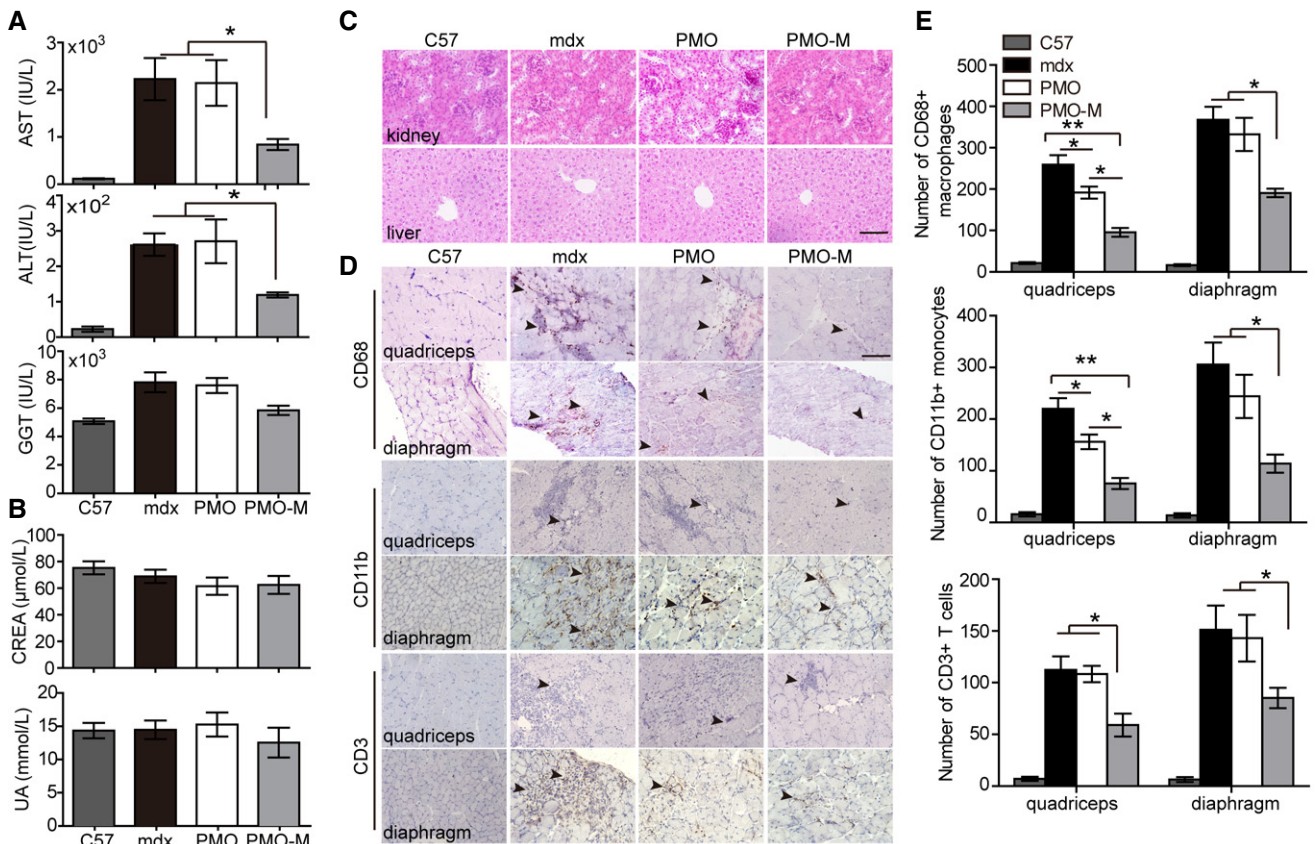

**Figure 6.  Evaluation of toxicology and inflammation in *mdx* mice treated with long-term repeated doses of PMO-M or PMO.**

PMO-M was administered intravenously into adult *mdx* mice at the PMO dose of 12.5 mg/kg/week for 3 weeks followed by 12.5 mg/kg/month for 3 months and tissues / blood were harvested 2 weeks after last injection.

A, B    Measurement of serum indices including liver enzymes (AST, ALT, and GGT) (A) and kidney markers (CREA and UA) (B) from wild-type *C57BL/6* (C57) (n = 3), untreated *mdx* controls (mdx) (n = 3), and *mdx* mice treated with PMO-M (n = 4) or PMO alone (n = 3) to reflect liver and kidney functions (*P < 0.05, one-way ANOVA post hoc Student–Newman–Keuls test).

C    Morphological examination of liver and kidney from wild-type *C57BL/6* (C57), untreated *mdx* controls (mdx), and treated *mdx* mice (scale bar: 100 μm).

D, E    Immunohistochemistry (D) and quantification (E) of macrophages, T cells, and monocytes in quadriceps and diaphragm from wild-type *C57BL/6* (C57) (n = 3), untreated *mdx* controls (mdx) (n = 3), and *mdx* mice treated with PMO-M (n = 4) or PMO (n = 3) (scale bar: 100 μm). The arrowheads point to CD68[+] macrophages, CD3[+] T cells, or CD11b[+] monocytes (*P < 0.05, **P < 0.001, one-way ANOVA post hoc Student–Newman–Keuls test).

Data information: Data were presented as mean ± sem. Exact P values are specified in Appendix Table S1.

## Discussion

Insufficient systemic delivery hinders accelerated clinical application of AO-mediated exon-skipping therapies for DMD. In this study, we demonstrated that an endogenous mitochondria-derived peptide, MOTS-c, enables enhanced delivery and uptake of PMO by promoting ATP production in energy-deficient dystrophic muscles. Importantly, MOTS-c showed an intrinsic muscle-targeting property as reflected by the accumulation in peripheral muscles of *mdx* mice after systemic intravenous injection. Long-term repeated administration of PMO-M at an extremely low dose of PMO (12.5 mg/kg) induced therapeutic levels of dystrophin restoration in peripheral muscles, with up to 25-fold increase in diaphragm, resulting in phenotypic rescue of *mdx* mice without any detectable toxicity. Our data demonstrate that MOTS-c serves as an adjuvant for AO-based exon-skipping therapeutics in DMD and thus might have therapeutic implications for other energy-deficient muscular diseases.

As MOTS-c primarily targets at skeletal muscles (Lee *et al*, 2015), we first tested its effect on muscle cells. Strikingly, MOTS-c augmented glycolytic rates in muscle cells and the enhancement was more dramatic in energy-deficient dystrophic muscle cells than normal muscle cells, with significant elevation of ATP levels. Considering DMD patients manifest impaired glucose tolerance and insulin resistance (Bostock *et al*, 2018), we speculated that it is likely MOTS-c facilitated glucose utilization to a much greater extent in dystrophic muscle cells than normal controls and thus generated more ATP, which resulted in enhanced PMO uptake without any direct interaction with PMO. Consistent with previous findings (Lee *et al*, 2015), our tissue distribution data revealed a substantial amount of MOTS-c accumulated in peripheral muscles but not in the heart. As the heart is a difficult organ to deliver to as our group has previously discovered (Gao *et al*, 2014; Han *et al*, 2016; Lin *et al*, 2020), it was unsurprising to see that PMO-M had a much less impact on the heart. However, it is also possible that the low dose

of PMO employed in the current study accounted for the limited effect observed in cardiac muscles as much higher doses were required to induce therapeutic levels of dystrophin expression in dystrophic hearts (Wu *et al*, 2011). Nevertheless, further studies on the combination of MOTS-c with higher doses of PMO or PMO-M with other drugs targeting at hearts are warranted.

In our previous study, we showed that glycine potentiates PMO activity by promoting muscle regeneration and thus resulting in enhanced incorporation of PMO in regenerating myofibers (Lin *et al*, 2020). Although MOTS-c functions via a different mechanism, we were curious about whether MOTS-c shows any better effect than glycine. Thus, we intravenously administered PMO-M or PMO in glycine (PMO-G) without additional administration of glycine at the dose of 25 mg/kg/week (PMO) for 3 weeks in adult *mdx* mice. The results showed that PMO-M induced significantly higher levels of dystrophin restoration than PMO-G (Appendix Fig S3A and B), suggesting that MOTS-c is more potent than glycine in enhancing PMO uptake. However, MOTS-c also targets the folate cycle and depletes intracellular 5-Methyltetrahydrofolate (5Me-THF) (Lee *et al*, 2015), a form of one-carbon unit donor required for cell division (Huang *et al*, 1999), extra caution needs to be taken in MOTS-c dosing when used in DMD patients, who manifest progressive muscle wasting (Sun *et al*, 2020). In our present study, we noticed a significantly reduced body-weight gain when additional MOTS-c was used in combination with PMO-M in adult *mdx* mice (Fig EV5). Since MOTS-c was shown to regulate homeostasis including muscle, adipose and plasma metabolites (Lee *et al*, 2015; Kim *et al*, 2019; Lu *et al*, 2019), we also examined the effect of MOTS-c alone on muscle pathologies, inflammations and body weight in *mdx* mice. Strikingly, MOTS-c alleviated muscle pathologies and inflammation with reduced body-weight gain at the dosing regimen of 500 μg per week for 3 weeks, followed by 500 μg per month for 3 months (Appendix Fig S4A–E), indicating that the appropriate titration of MOTS-c is beneficial for DMD. Nevertheless, a detailed study on the optimal titration of MOTS-c is warranted prior to its clinical deployment.

Overall, our study demonstrates that MOTS-c enables augmented glycolytic flux and energy production in energy-deficient dystrophic muscles. Co-administration of MOTS-c and low doses of PMO significantly enhanced PMO uptake and activity in dystrophic muscles and elicited therapeutic levels of dystrophin restoration and phenotypic rescue without any detectable toxicity. Our data provide evidence for the first time that MOTS-c can be used as a delivery vehicle for AOs and thus accelerate the clinical deployment of PMO for DMD.

# Materials and Methods

### Animals and injections

Adult *mdx* (6–8 weeks old) and age-matched *C57BL/6* mice were used in all experiments (numbers of mice per group were specified in Figure legends). The mice were housed under specific pathogen-free conditions in a temperature-controlled room. The experiments were carried out in the Animal unit, Tianjin Medical University (Tianjin, China), according to procedures authorized by the institutional ethical committee (Permit Number: 2019-0004). For local intramuscular injection, 0.5 μg PMO was dissolved in saline with or without MOTS-c (20 μg) and the total volume was 40 μl. For intravenous injections, various amounts of PMO with MOTS-c (500 μg) or GF (5%) or glycine (5%) were injected into tail vein of *mdx* mice at the dose of 12.5, 25 or 50 mg/kg per week, respectively, for 3 weeks and the total volume was 120 μl. For the long-term systemic study, MOTS-c (500 μg) with PMO in saline was intravenously injected into *mdx* mice at the dose of 12.5 mg/kg per week for 3 weeks followed by 12.5 mg/kg per month for 3 months. To examine the systemic activity of MOTS-c alone, we intravenously injected MOTS-c into adult *mdx* mice at the dose of 500 μg per week for 3 weeks followed by 500 μg per month for 3 months. Mice were killed by $CO_2$ inhalation 2 weeks after last injection unless otherwise specified, and muscles and other tissues were snap-frozen in liquid nitrogen-cooled isopentane and stored at −80°C.

### Oligonucleotides and peptides

FITC-labeled and unlabeled PMO were synthesized and purified by GeneTools (Corvallis, OR, US). PMO (5'-GGCCAAACCTCGGCT TACCTGAAAT-3') sequence was targeted to murine dystrophin exon 23/ intron 23 boundary sites as reported previously (Harding *et al*, 2007). Rhodamine B-labeled and unlabeled MOTS-c peptides (MRWQEMGYIFYPRKLR; Lee *et al*, 2015) were synthesized and purified to > 98% purity by ChinaPeptides Co. Ltd (Suzhou, China).

### Extracellular acidification rate (ECAR) assay

Conditionally immortalized murine H2K and H2K-tsA58 *mdx* cells derived from *H-2K^b-tsA58* mice with a nonsense point mutation in exon23 of *dmd* gene were kindly provided by Professor Terry Partridge (Children's National Medical Center, Center for Genetic Medicine Research, Washington DC, USA) and cultured as previously reported (Morgan *et al*, 1994). Briefly, cells were grown in Dulbecco's modified Eagle's medium (DMEM) supplemented with 20% fetal calf serum (FBS), 2% chick embryo extract, 2% L-glutamine, 1% penicillin and streptomycin, and 20 U/ml mouse recombinant IFN-γ (Invitrogen, USA) at 33°C in 10% $CO_2$. H2K and H2K *mdx* cells ($2 \times 10^4$) were seeded in the XF-24 microplate at 33°C under 10% $CO_2$ overnight prior to the treatment with MOTS-c peptide (10 μM) for 72 h. Cells were stimulated with glucose to determine active glycolytic rate, with oligomycin to determine maximum glycolytic capacity, and with 2-DG to determine glycolytic capacity. The glycolytic rates were determined as per manufacturer's instructions (TeSlaa & Teitell, 2014) and presented as the percentage in relative to the basal level. All readings were normalized to total DNA content.

### Cellular uptake

H2K *mdx* cells ($5 \times 10^4$) were grown in DMEM supplemented with 20% FBS, 2% chick embryo extract, 2% L-glutamine, 1% penicillin and streptomycin, and 20 U/ml mouse recombinant IFN-γ (Invitrogen, US) at 33°C in 10% $CO_2$. For the detection of cellular uptake of PMO, H2K *mdx* cells ($5 \times 10^4$) were seeded in the Nunc Glass Base Dishes and cultured overnight, and FITC-labeled PMO (5 μM) mixed with MOTS-c (10 μM) were added into the cultured H2K *mdx* cells for 24 h. Cells were washed with cold phosphate-buffered saline (PBS) for 3 times to remove the non-specific bound PMO and the

nuclei were counterstained with DAPI (Invitrogen, USA). Images were obtained using a confocal fluorescence microscope (Olympus FV1000, Olympus, Japan). For the quantification, H2K *mdx* cells were seeded and cultured in the 96-well plate for 24 h. Subsequently, FITC-labeled PMO (500 nM) mixed with MOTS-c (10 μM) were added into H2K *mdx* cells and incubated for 24 h, followed by washing with PBS and lysed with RIPA lysis buffer (Beyotime, China). Subsequently, cell lysates were measured with the 96-well plate reader (Nunc, US) with the excitation wavelength of $488 \pm 20$ nm and the maximal emission wavelength of $520 \pm 20$ nm.

## ATP measurement

Murine H2K/H2K *mdx* cells (2000/well) were seeded and cultured in the 96-well plate at 33°C under 10% $CO_2$ overnight, followed by treatment with MOTS-c (10 μM) for 72 h. Cells were collected and re-suspended with 50 μl DPBS and 25 μl cell suspension were mixed with the reaction substrate (CellTiter-Glo Luminescent Cell Viability Assay kit, Promega, USA) and reacted for 20 min at room temperature. The Luminous value (EnSpire Multimode Plate Reader, PerkinElmer) was measured individually and reported as the percentage in relative to the basal level. For the measurement of ATP levels in muscle tissues, muscles were harvested and snap-frozen in liquid nitrogen, and 10-20 mg of 4- to 6-μm-thick cryosections were collected into a 1.5-ml Eppendorf tube. $HClO_4$ (0.4 M, 600 ml) was added into the tubes and mixed thoroughly to dissolve the sections. The tube was then centrifuged at 376 *g* in 4°C for 5 min, and the supernatant was transferred to a new tube. To fully dissolve the tissues, another 400 ml 0.4 M $HClO_4$ was added into the precipitate. Subsequently, the supernatant was used for ATP assay as per manufacturer's instructions (CellTiter-Glo Luminescent Cell Viability Assay kit, Promega, USA).

## Tissue distribution

For the distribution of MOTS-c, Rhodamine B-labeled MOTS-c (500 μg) was dissolved in 120 μl saline and injected intravenously into adult *mdx* mice, perfusion was performed 2 h after injection with 50 ml cold PBS, and body-wide tissues were harvested for imaging with IVIS spectrum (PerkinElmer, USA). To examine the bio-distribution of PMO, FITC-labeled PMO at the dose of 50 mg/kg was mixed with MOTS-c (500 μg) and were intravenously administered into adult *mdx* mice with a total volume of 120 μl for once. Mice were terminally anesthetized 48 h after injection and perfused with 50 ml of cold PBS to wash out free PMO in circulation. Body-wide muscles, liver, lung, spleen, and kidney were collected for imaging and quantified with the IVIS imaging system as per manufacturer's instructions. For the quantification of fluorescence intensity in each individual tissue, each tissue was encircled as the interested region and the fluorescence intensity was calculated automatically with the living Image® software (caliper life science, USA).

## Immunohistochemistry and histology

Series of 8 μm sections from tibialis anterior (TA), quadriceps, gastrocnemius, triceps, abdominal, diaphragm, and hearts were examined for dystrophin expression with a rabbit polyclonal

antibody P7 (1:100; Fairway Biotech, England). The P7 antibody binds to the rod domain (exon 57) of the dystrophin protein. The primary antibody was detected by goat-anti-rabbit IgG Alexa Fluor 594 (Molecular Probe, UK). For the quantification of dystrophin-positive fibers, the maximum number of dystrophin-positive fibers in one section was counted using the Zeiss AxioVision fluorescence microscope (Zeiss, Germany) and the muscle fibers were defined as dystrophin-positive when more than two thirds of the single fiber showed continuous staining as described previously (Yin *et al*, 2010). The serial sections were stained with a panel of polyclonal and monoclonal antibodies for the detection of DAPC components. Rabbit polyclonal antibody to neuronal nitric oxide synthase (1:200), mouse monoclonal antibodies to β-dystroglycan and α-sarcoglycan were used as per manufacturer's instructions (1:200, Novocastra, UK). The primary antibodies were detected with goat-anti-rabbit IgG Alexa Fluor 488 or goat-anti-mouse IgGs Alexa 488 (Molecular Probe, UK). The M.O.M. blocking kit (Vector Laboratories Inc., USA) was applied for the immunostaining of DAPC as per manufacturer's instructions. Muscle fiber types were stained with the following primary antibodies: mouse IgG2b monoclonal anti-type I MHC (clone BA-D5, 1:100), mouse IgG1 monoclonal anti-type IIa MHC (clone SC-71, 1:100), mouse IgM monoclonal anti-type IIb MHC (clone BF-F3, 1:10) (provided by the Developmental Studies Hybridoma Bank at the University of Iowa, USA), followed by detection with secondary antibodies including goat-anti-mouse IgG Fc IIb Alexa Fluor 488, goat-anti-mouse IgG Fc I Alexa Fluro 633 and goat-anti -mouse IgM Alexa Fluor 594 (1:200, Molecular Probe, UK), respectively, as described previously (Vila *et al*, 2015). To examine the presence of macrophages, T cells, and monocytes in muscle tissues from treated *mdx* or control mice, muscle tissues were fixed in Bouin's solution (Sigma, USA) and embedded with paraffin. CD68[+] macrophages, CD3[+] T cells and CD11b[+] monocytes were stained with rabbit polyclonal antibodies: CD68 (1:400, Abcam, UK), CD3 (1:400, Abcam, UK) and CD11b (1:500, Bioss, China), respectively, and detected by goat-anti-rabbit secondary antibody (Sigma, USA). To measure the fibrotic areas, Masson's trichrome staining kit (Sigma, USA) was applied as per manufacturer's instructions. Routine H&E staining was used to examine the overall liver, kidney, and muscle morphology and assess the level of infiltrating mononuclear cells.

## Centrally nucleated fibers (CNFs) and cross-sectional area (CSA) measurement

Quadriceps and triceps muscles from *mdx* mice treated with long-term repeated doses of PMO-M or PMO were examined. 500–1,000 dystrophin-positive fibers for each tissue sample were randomly chosen, counted, and assessed for the presence of central nuclei using a Zeiss AxioVision fluorescence microscope (Zeiss, Germany). Fibers with one or more nuclei were not located at the periphery of the fiber were defined as centrally nucleated. For the measurement of cross-sectional areas, sections (8 μm) were cut at mid-belly of triceps from *mdx* mice treated with long-term repeated doses of PMO-M or PMO and stained with hematoxylin and eosin and images were acquired with conventional microscope (OLYPUS BX51, Japan) at five different fields randomly at 100× magnification. The interactive measurements tool MetaMorph (Molecular device, US) was used to process the images to measure the cross-sectional area,

historical parameters of the total fibers in each section were calculated and analyzed as described previously (Ran *et al*, 2020).

## Protein extraction and western blot

The collected sections were placed in a 1.5-ml polypropylene Eppendorf tube on dry ice and lysed with 150 ml protein extraction buffer containing 125 mM Tris–HCl pH 6.8, 10% SDS, 2 M urea, 20% glycerol and 5% 2-mercaptoethanol. The mixture was boiled for 5 min, followed by centrifugation at 13,523 $g$ for 10 min at 4°C, and the supernatant was collected for protein measurement with the Bradford assay (Sigma, USA). Various amounts of protein from normal *C57BL/6* TA muscles as a positive control and from muscles of treated or untreated *mdx* mice were loaded onto SDS–PAGE gels (4% stacking and 6% resolving). Samples were electrophoresed for 4 h at 80 V and transferred to polyvinylidene fluoride (PVDF) membrane overnight at 110 mA at 4°C. The membrane was then washed and blocked with 5% skimmed milk and probed with DYS1 (monoclonal antibody against dystrophin R8 repeat, 1:200, Novocastra, UK) overnight. Quantification is based on the band intensity and area with ImageJ software, and compared with that from *C57BL/6* TA muscles. Briefly, the densitometric intensity of each band, including dystrophin (the area of the major dystrophin band) and α-actinin, was measured. The band area was kept constant between lanes for an individual blot for analysis. The dystrophin values were divided by their respective α-actinin values. The dystrophin /α-actinin ratios of treated samples were normalized to the average *C57BL/6* dystrophin /α-actinin ratios (from serial dilutions).

## ELISA for PMO in tissues

ELISA was used to detect the amount of PMO in muscle tissues as described previously (Han *et al*, 2018). Briefly, a DNA probe was designed with sequences complementary to PMO (synthesized by The Beijing Genomics Institute, Beijing, China) as follows: 5'-**ATTT-CAGG**TAAGCCGAGG**TTTGGCC**-3' (bold means phosphorothioated). The 5' and 3' ends of the probe were labeled with digoxigenin and biotin, respectively. Standard PMO samples and muscle tissues (100 mg/ml) were digested with 20 mg/ml proteinase K at 55°C overnight. After PMO-probe hybridization, the avidin-biotin interaction of the hybridized probe was performed on Pierce NeutrAvidin Coated 96-well plates (Thermo Fisher Scientific, MA, USA). Unhybridized probes were digested with micrococcal nuclease at 10 U/µl (Thermo Fisher Scientific, MA, USA). Then the hybridized probes were reacted with rabbit monoclonal antibody (1:1,000; Cell Signaling Technology, MA, USA) to digoxigenin, followed by detection with peroxidase-conjugated goat anti-rabbit IgG (Abcam, Cambridge, UK). Signals from the PMO-hybridized probe were detected at 450 nm with TME Substrate (Solarbio, Beijing, China) in a monochromator EnSpire Multimode plate reader (PerkinElmer, Boston, MA, USA).

## Functional grip strength and running wheel test

Treated and control mice were tested using a commercial grip strength monitor (Chatillon, West Sussex, UK) as described previously (Lin *et al*, 2020). Briefly, each mouse was held 2 cm from the base of the tail, allowed to grip a protruding metal triangle

### The paper explained

**Problem**

Antisense oligonucleotide (AO)-mediated exon-skipping therapy shows promise in Duchenne muscular dystrophy (DMD), a devastating muscle wasting condition, with three AO drugs approved. However, insufficient systemic delivery remains a major hurdle to affordable and efficacious clinical application.

**Results**

Our study demonstrated that MOTS-c, a mitochondria-derived bioactive natural peptide, can augment glycolytic flux and energy production capacity of dystrophic muscles. This in turn results in enhanced AO uptake and activity and improved muscle function and outcomes in dystrophic mice.

**Impact**

Our results suggest that MOTS-c enables enhanced AO uptake and activity in dystrophic muscles by supplementing cellular energy stores and may have therapeutic implications for exon-skipping therapeutics in DMD and other energy-deficient disorders.

bar attached to the apparatus with the forepaws and pulled gently until the mice released the grip. The force exerted was recorded and five sequential tests were carried out for each mouse, averaged at 30 s apart. Subsequently, the readings for force recovery were normalized by the body weight. For muscle endurance, treated and control mice were placed in cages equipped with voluntary running wheels (Zhenhua, Anhui, China) and monitored for 48 h in a quiet and cleanly room at a temperature of 24 ± 1°C. Running circles were recorded and distance was analyzed.

## Serum enzyme measurement

Mouse blood was taken immediately after cervical dislocation and centrifuged at 211 $g$ for 30 min and stored at −80°C. Analysis of serum creatine kinase (CK), aspartate transaminase (AST), alanine aminotransferase (ALT), and creatinine (CREA) was performed in School of laboratory, Tianjin Medical University (Tianjin, China). Gamma Glutamyl Transferase (GGT) Assay Kit (G0434W, Grace Biotechnology, China) was applied to measure the level of serum GGT as per manufacturer's instructions.

## Data analysis

All data are reported as mean values ± SEM. Statistical differences between different treated groups were evaluated by Sigma Stat (Systat Software Inc., Chicago, IL, USA). Both parametric and non-parametric analyses were applied as specified in figure legends. Sample size was determined by G*Power 3.1.7 (Power analysis and Sample size). Significance was determined based in $P < 0.05$. Library (basic trendline) based on R version 3.6.3 was used to analyze the correlation between the level of dystrophin expression and distribution of PMO in different muscles of *mdx* mice. Level of confidence interval used is 0.95 by default. For animal studies, age-matched mice were used and randomly divided into different groups. The investigators were not blinded to the group allocation during data collection and/

or data analysis because all samples were analyzed in the same way.

## Data availability

This study includes no data deposited in external repositories.

**Expanded View** for this article is available online.

## Acknowledgements

The authors acknowledge Dr Yiqi Seow (Molecular Engineering Laboratory, A*STAR, Singapore) for critical review of the manuscript. This study was funded by National Key R&D Program of China (Grant No.2017YFC1001902), National Natural Science Foundation of China (Grant No. 82030054, 81672124, and 81802124); Tianjin Research Innovation Project for Postgraduate Students (YJSCX201802), and Tianjin Municipal 13th five-year plan (Tianjin Medical University Talent Project).

## Author contributions

HY and NR conceived the project, designed the experiments, and analyzed the data. HY supervised the project, provided the funding, interpreted the results, and wrote the manuscript with input from all authors. NR performed all the *in vitro* experiments, *in vivo* imaging of MOTS-c and PMO bio-distribution, and analyzed the data. CL performed the injection, WB, grip strength and running wheel measurement, and helped with the data analysis. GH performed serum enzyme assay. LL, MG, and YW helped with animal experiments and tissue harvesting. SB and HMM reviewed the data and provided advice.

## Conflict of interest

The authors declare that they have no conflict of interest.

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
