## [Review Process File · EMBO Molecular Medicine]

MOTS-c promotes phosphorodiamidate morpholino oligomer uptake and efficacy in dystrophic mice

Ning Ran, Caorui Lin, Ling Leng, Gang Han, Mengyuan Geng, Yingjie Wu, Scott Bittner, Hong Moulton, and HAIFANG YIN

DOI: [10.15252/emmm.202012993](https://doi.org/10.15252/emmm.202012993)

Corresponding author: HAIFANG YIN (haifangyin@tmu.edu.cn)

Review Timeline:

Submission Date:	23rd Jun 20
Editorial Decision:	24th Jul 20
Revision Received:	13th Oct 20
Editorial Decision:	9th Nov 20
Revision Received:	16th Nov 20
Accepted:	18th Nov 20

Editor: Zeljko Durdevic

Transaction Report:

24th Jul 2020

Dear Prof. Yin,

Thank you for the submission of your manuscript to EMBO Molecular Medicine. We have now received feedback from the three reviewers who agreed to evaluate your manuscript. As you will see from the reports below, the referees acknowledge the interest of the study but also raise serious concerns that should be addressed in a major revision.

Addressing the reviewers' concerns in full will be necessary for further considering the manuscript in our journal, and acceptance of the manuscript will entail a second round of review. EMBO Molecular Medicine encourages a single round of revision only and therefore, acceptance or rejection of the manuscript will depend on the completeness of your responses included in the next, final version of the manuscript. For this reason, and to save you from any frustrations in the end, I would strongly advise against returning an incomplete revision.

We realize that the current situation is exceptional on the account of the COVID-19/SARS-CoV-2 pandemic. Therefore, please let us know if you need more than three months to revise the manuscript.

I look forward to receiving your revised manuscript.

***** Reviewer's comments *****

Referee #1 (Remarks for Author):

The paper by Ran et al. and entitled "MOTS c promotes phosphorodiamidate morpholino oligomer uptake and efficacy in dystrophic mice" aims to determine whether PMO uptake and activity would be improved if PMOs are used in combination with MOTS-c in dystrophic muscle cells or DMD mice. This is a very interesting paper rich in data. However, some corrections/improvements need to be done before being considered for publication.

Major points

A recurrent point is that for clarity, the authors should specify in each figure's legend how long after

the treatment the analyses were done: 48h, etc... as it is not always easy to identify the different time points used by the authors. Some clarity regarding the methods used is needed (see below). Also, the authors should investigate whether the efficacy varies between different fibre types and between fibres with different metabolic status (fibre type I v Type IIb, rich v poor in mitochondria)? The authors should also quantify the images to support their conclusion (see below).

1- MOTS-c triggers increased energy production in dystrophic muscles & Figure 1

What is the technical reason to justify n=3 for the percentage of ECAR exchange in Fig1b and n=6 in Figure 1a & c?

2- MOTS-c promotes PMO uptake in dystrophic muscles in vitro and in vivo & Figure 2 & supplementary Figure 1

Figure 2b and C are obtained 48h after injection and show the level of PMO in different muscles/tissues.

- What happen in TA muscles (TA data being shown in Figure 1)?

- Transferring supplementary Figure 2a-c in the main Figure 2 would highlight the efficacy of PMO-M in rescuing dystrophin expression. In Supp Figure 2, an immunostaining of the TA is shown. Thus to be consistent, the authors should also add the PMO-M level in TA muscle at 48h post-injection in Figure 2 b.

- Does the dystrophin expression level correlate with the PMO-M level observed in different muscle at 48h post-injection and/or with MOTS-c distribution?

- Is the efficacy depending on the fibre types? Does the Soleus show a lower or higher efficacy than what is observed in TA or Q or G muscles?

Figure 2c: How is the MFI normalized?

Figure 2d: The percentage of fibres expressing dystrophin across muscle section is missing.

Similarly, in supplementary figure 1a, the number of fibres expressing dystrophin across muscle section for the different conditions should be given.

3- MOTS-c enhances PMO activity in mdx mice in a saturable manner & Figure 3 & Supplementary figure 2 & supplementary Figure 3

- Transferring supplementary Figure 2a-c in the main Figure 2 would highlight the efficacy of PMO-M in rescuing dystrophin expression.

- Figure 3e: the number of fibres expressing dystrophin across muscle section for the different conditions should be given.

- Same question as in above: Is the efficacy varying with the fibre types?

- Figure 3g- fold change graph: SEM bars are missing

- Supplementary Figure 3: the authors should briefly explain in the main text why comparing PMO-M with PMO-GF to allow non-expert readers understand the rational.

4- PMO-M induces long-term therapeutic efficacy and phenotypic rescue in mdx mice & Figure 4

- Figure 4: Same question as above: Is the efficacy varying with the fibre types?

- Figure 4b: Percentage of fibres positive for dystrophin should be given

- Figure 4d-fold change: error bars are missing

- Figure 4 b-d and f: How many weeks after the treatment these analyses were done?

5- PMO-M improves muscle pathologies without detectable toxicity in mdx mice & Figure 5

- Figure 5b: the pictures for PMO-M alpha should be centred the same way to show the rescue in dystrophin, alpha sarcoglycan, B dystroglycan and nNOS.

- Figure 5d: The authors should detail a little bit more in the legend the different parameters measured.
- Figure 5f: to be objective regarding the inflammatory response, quantification of CD68 should be done in the 4 groups. To be more complete other markers should be measured: eg CD4, CD8, CD11b.
- Supplementary Figure 6c: Quantification of CD68 should be done as well to support the conclusion.

6- Methods:

The method section could be improved. The authors should specify in the legend of the figures which cell lines were used: C2C12, H2K WT and mdx? Which H2K mdx cell lines was used (mutation exon 23, 52, else?), What is the PMO sequence used?

- H2K mdx and WT cell lines: where these cell lines come from? Gift or bought? Which mutation H2K mdx carry? Which background of mice the cells have been generated on? Protocol for expansion and maintenance is incomplete.
- Why is there a paragraph on C2C12 cell line for PMO uptake while all the results are shown on dystrophic cells (so presumably on H2K-mdx cell lines)?
- Muscle endurance: for how long the mice were monitored in cage and when after the treatment?

Minor points:

- Page 4 : define GF
- A space before the references in the text is often missing.
- Figure 1A: same scale for the Y-axes should be used for H2K and H2K-mdx graphs to be able to compare them directly.
- Page 5, the word anaerobic should be added in sentence as follow: "This result demonstrates that the increase in ECAR upon addition of glucose is due to anaerobic glycolysis rather than other sources."

Referee #2 (Comments on Novelty/Model System for Author):

Ran et al examined the effects of a peptide MOTS-c on restoring dystrophin expression with exon skipping. This peptide has shown to enhance glucose utilization and ATP production, thereby contributing to energy production.

The authors demonstrated improved uptake of PMO along with the peptide MOTS-c in the skeletal muscles of the mdx mouse model. They observed the long-term effects of the treatment at repeated low doses of PMO and the peptide without apparent toxicity. Additionally, they further supported their study by comparing the efficacy of PMO + MOTS-c with PMO in GF (glucose-fructose) and PMO in glycine, in which they found that the former was superior in restoring dystrophin expression.

Referee #2 (Remarks for Author):

1. In the introduction section, in addition to eteplirsen, golodirsen is another PMO antisense oligonucleotide that has been approved by the FDA, and viltolarsen was approved in Japan. These need to be mentioned.
2. The authors used rhodamine B-labeled MOTS-c to quantify the levels in tissues but did not

include TA muscles in Figure 2. It would be interesting to add TA here so that the correlation between the uptake and dystrophin levels can be examined.

3. In the introduction section, the authors state that "repeated administration of MOTS c with extremely low doses of PMO (12.5mg/kg) elicited therapeutic levels of dystrophin restoration and functional improvements in mdx mice without any adverse effect ."; however, this is an overstatement as adverse effects were not examined extensively.

4. the authors claim "uniform distribution of dystrophin-positive myofibers" (page 7) and "uniform expression of dystrophin" (page 8); however, these are confusing as the distribution appears to be mosaic rather than uniform in IHC.

5. The authors examined AST and ALT levels; however, GGT (gamma-GTP) is more appropriate to examine liver function as AST and ALT levels are already elevated in dystrophic models and can mask the toxic effects.

6. As authors employed peptides, immune response (e.g. CD3, CD4, CD8) should be examined.

7. No western blotting included for the heart muscle. It can be incorporated to see if there is even a slight fold change in the dystrophin level in studying the long-term therapeutic efficacy because cardiomyopathy is predominantly prevalent in DMD patients.

8. The authors need to mention the condition for the toxicity studies in figure 5 legend, as it is confusing to follow. (i.e. the timeline of the administration, concentration, etc.) Is it the long-term study at the lower dose of PMO?

9. Quantification and analysis of the myofibers (cross-section area, number of centrally nucleated fibers) are required in addition to the H and E staining of the diaphragm and quadriceps for the long-term study.

10. As the authors mentioned in the study that MOTS-c depletes intracellular 5-Methyltetrahydrofolate (5Me-THF) which is essential for cell division, a subsequent study might be necessary to determine the tolerated dose which doesn't lead to muscle wasting.

11. The sequence of the peptide and PMO need to be mentioned in the material-methods section.

Reviewer #1

Major points: A recurrent point is that for clarity, the authors should specify in each figure's legend how long after the treatment the analyses were done: 48h, etc... as it is not always easy to identify the different time points used by the authors. Some clarity regarding the methods used is needed (see below). Also, the authors should investigate whether the efficacy varies between different fibre types and between fibres with different metabolic status (fibre type I v Type IIb, rich v poor in mitochondria)? The authors should also quantify the images to support their conclusion (see below).

Response: We thank Reviewer#1 for these points and have addressed these points individually as detailed below.

Point 1: MOTS-c triggers increased energy production in dystrophic muscles & Figure 1 What is the technical reason to justify n=3 for the percentage of ECAR exchange in Fig1b and n=6 in Figure 1a & c?

Response: We thank Reviewer#1 for pointing this error out. Basically, n=3 means three time-points in the glycolysis stage (30min, 40min and 50min) as the ECAR change in Figure 1B was derived from mean values of ECAR of MOTS-c-treated H2K *mdx* cells relative to untreated cells at the same 3 time-points (30min, 40min and 50min) (Figure 1A). The same applies to H2K cells. Therefore, the sample number was 6 and we have rectified the error in the corresponding Figure legends.

Point 2: MOTS-c promotes PMO uptake in dystrophic muscles in vitro and in vivo & Figure 2 & supplementary Figure 1. Figure 2b and C are obtained 48h after injection and show the level of PMO in different muscles/tissues. What happen in TA muscles (TA data being shown in Figure 1)?

Response: We thank Reviewer#1 for pointing this out and have added TA muscles in Figure 2B and 2C as Reviewer#1 has suggested. Based on the data, there was a marginal increase in the level of fluorescence intensity in TA muscles treated with PMO-M compared to PMO. Consistent with previous observations (Lin et al. *Molecular Therapy* (2020) 28(5):1339-1358; Han et al. *Nature Communications* (2016) 7:10981), the fluorescence intensity was relatively lower in TA muscles than quadriceps (Q), gastrocnemius (G), triceps (T) and abdominal muscles (A), which is likely due to the smaller volume of TA muscles. Quantification of PMO in TA muscles and gastrocnemius confirmed significantly increased uptake of PMO in peripheral muscles from PMO-M-treated *mdx* mice compared to PMO-treated *mdx* mice (Figure EV1). We have provided the data as Figure EV1.

Transferring supplementary Figure 2a-c in the main Figure 2 would highlight the efficacy of PMO-M in rescuing dystrophin expression. In Supp Figure 2, an immunostaining of the TA is shown. Thus to be consistent, the authors should also add the PMO-M level in TA muscle at 48h post-injection in Figure 2b.

Response: We thank Reviewer#1 for the helpful suggestion and have transferred Supplementary Figure 2 to the main Figure 2 as Figure 2F-H as Reviewer#1 has recommended. Also we have added TA muscles in Figure 2B and 2C as noted above.

Does the dystrophin expression level correlate with the PMO-M level observed in different muscle at 48h post-injection and/or with MOTS-c distribution?

Response: To make it clearer, we were unable to detect any dystrophin expression in peripheral muscles of treated *mdx* mice merely 48hrs after single intravenous injection of PMO at 50mg/kg as demonstrated below (Figure 1) as the optimal time for examining dystrophin expression is 10-14 days after injection as shown by previous studies (Lu et al., *Nature Medicine* (2003)9:1009-1014; Lu et al., *PNAS* (2005) 102:198-203; Alter et al., *Nature Medicine* (2006)12(2):175-177). Nevertheless, a correlation was established between the level of dystrophin expression in different muscles of *mdx* mice treated with PMO-M intravenously at the PMO dose of 50mg/kg/week for 3 weeks and tissues were examined two weeks after last injection (Figure 2H) and distribution of PMO-M (Figure 2C; 48hrs after single intravenous injection of FITC-labeled PMO (50mg/kg)) and / or MOTS-c (Figure 1D; 2hrs after single intravenous of Rhodamine B-labeled MOTS-c (500µg)) in different muscles of treated *mdx* mice.

Figure 1. Western blot to detect dystrophin expression in peripheral muscles of *mdx* mice treated with PMO-M or PMO alone intravenously at single dose of 50mg/kg and tissues were harvested 48 hours after injection. 2 µg and 1 µg total protein from *C57BL/6* and 100 µg from muscle samples of untreated and treated *mdx* mice were loaded. α-actinin was used as the loading control.

Is the efficacy depending on the fibre types? Does the Soleus show a lower or higher efficacy than what is observed in TA or Q or G muscles?

Response: We are grateful for Reviewer#1's questions. To make it clearer, MOTS-c primarily contributes to energy production by enhancing glycolytic flux as identified and reported by Lee *et al.* (*Cell Metabolism* (2015):21:443-454). Therefore, we assume that MOTS-c would not have a direct impact on mitochondria in terms of energy production, which is the downstream of glycolysis. Nevertheless, we have examined the effect of PMO-M on extensor digitorum longus (EDL), soleus and triceps representing fast-twitch, slow-twitch and mixed fiber types, respectively, from *mdx* mice treated with PMO-M at the PMO dose of 12.5mg/kg/week for 3 weeks and tissues were harvested two weeks after last injection as we were unable to detect any dystrophin expression in muscles from *mdx* mice treated with single dose of PMO-M at the PMO dose of 50mg/kg and tissues were harvested 48 hrs later as shown in Figure 1 (stated above). The results showed significant increases in the number of dystrophin-positive fibres in different muscles from PMO-M-treated *mdx* mice compared to PMO-treated *mdx* mice irrespective of fibre types, however, the fold change was much greater in type IIb myosin heavy chain (MHC)-positive fast-twitch EDL muscles than type I MHC-positive slow-twitch soleus muscles, confirming that glycolysis is primarily responsible for energy production, resulting in higher PMO uptake in fast-twitch fibres. We have presented the data as Figure EV3A and EV3B as Reviewer#1 has recommended.

Figure 2c: How is the MFI normalized?

Response: To make it clearer, the MFI was normalized by deducting background fluorescence as judged by the imaging software. For the quantification of fluorescence intensity in each individual tissue, each tissue was encircled as the interested region and the fluorescence intensity was calculated automatically with the living Image® software (caliper life science, US). MFI represents the mean value of 3 (Figure 2C) or 6 (Figure 1D) biological samples.

Figure 2d: The percentage of fibres expressing dystrophin across muscle section is missing. Similarly, in supplementary figure 1a, the number of fibres expressing dystrophin across muscle section for the different conditions should be given.

Response: We have provided the quantitative data of dystrophin-positive fibres in Figure 2D and Supplementary Figure 1A (we have renamed the Figure as Figure EV2B per EMBO Molecular Medicine's instruction) as Reviewer#1 has suggested.

Point 3: MOTS-c enhances PMO activity in *mdx* mice in a saturable manner & Figure 3 & Supplementary figure 2 & supplementary Figure 3. Transferring supplementary Figure 2a-c in the main Figure 2 would highlight the efficacy of PMO-M in rescuing dystrophin expression.

Response: We thank Reviewer#1 for the helpful suggestion and have added Supplementary Figure 2 in Figure 2 as Figure 2G-I as Reviewer#1 has recommended.

Figure 3e: the number of fibres expressing dystrophin across muscle section for the different conditions should be given.

Response: We have provided the quantitative data of dystrophin-positive fibres in Figure 3E as Reviewer#1 has suggested.

Same question as in above: Is the efficacy varying with the fibre types?

Response: We have examined the effect of PMO-M on extensor digitorum longus (EDL), soleus and triceps representing fast-twitch, slow-twitch and mixed fiber types, respectively, from *mdx* mice treated with PMO-M at the PMO dose of 12.5mg/kg/week for 3 weeks and tissues were harvested two weeks after last injection. The results showed significant increases in the number of dystrophin-positive fibres in different muscles from PMO-M-treated *mdx* mice compared to PMO-treated *mdx* mice irrespective of fibre types, however the fold change was much greater in fast-twitch fibres e.g. EDL than slow-twitch fibre (soleus), confirming that glycolysis is primarily responsible for energy production, resulting in higher PMO uptake in fast-twitch fibres. We have provided the data as Figure EV3A and EV3B.

Figure 3g- fold change graph: SEM bars are missing

Response: To make it clearer, Figure 3G showed the fold change of PMO-M relative to PMO, in which the mean value of PMO-M and PMO was used for a fair comparison. Therefore, no error bar was presented.

Supplementary Figure 3: the authors should briefly explain in the main text why comparing PMO-M with PMO-GF to allow non-expert readers understand the rationale.

Response: We thank Reviewer#1 for the helpful suggestion and have explained the rationale behind for the comparison between MOTS-c and GF as Reviewer#1 has suggested. Now it reads as follows: “As GF was shown to enhance PMO uptake in dystrophic muscles by replenishing energy stores (Han et al., 2016), we wondered whether MOTS-c would outperform GF in restoring dystrophin expression. In comparison with PMO in GF (PMO-GF), PMO-M....”

Point 4: PMO-M induces long-term therapeutic efficacy and phenotypic rescue in *mdx* mice & Figure 4. Figure 4: Same question as above: Is the efficacy varying with the fibre types?

Response: Again we thank Reviewer#1 for this question and have examined the representative muscles including fast-twitch extensor digitorum longus (EDL) and slow-twitch soleus from *mdx* mice treated with long-term repeated injections of PMO and PMO-M as Reviewer#1 has suggested. The results showed significant increases in the number of dystrophin-positive fibres in different muscles from PMO-M-treated *mdx* mice compared to PMO-treated *mdx* mice irrespective of fibre types, however the fold change was much greater in fast-twitch fibres e.g. EDL than slow-twitch fibre (soleus), confirming that glycolysis is primarily responsible for energy production, resulting in higher PMO uptake in fast-twitch fibres. And we have presented the data as Figure EV4A and EV4B.

Figure 4b: Percentage of fibres positive for dystrophin should be given

Response: We have provided the quantitative data for dystrophin-positive fibres in Figure 4B as Reviewer#1 has recommended.

Figure 4d-fold change: error bars are missing

Response: To make it clearer, Figure 4D showed the fold change of PMO-M relative to PMO, in which the mean value of PMO-M and PMO was used for a fair comparison. Therefore, no error bar was presented.

Figure 4 b-d and f: How many weeks after the treatment these analyses were done?

Response: The tissues were harvested and examined two weeks after last injection as illustrated in Figure 4A.

Point 5: PMO-M improves muscle pathologies without detectable toxicity in mdx mice & Figure 5. Figure 5b: the pictures for PMO-M alpha should be centred the same way to show the rescue in dystrophin, alpha sarcoglycan, B dystroglycan and nNOS.

Response: We thank Reviewer#1 for the comments and have re-stained the samples for PMO-M and re-organized the pictures in Figure 5B to make sure the arrow was placed in the center as Reviewer#1 has recommended.

Figure 5d: The authors should detail a little bit more in the legend the different parameters measured.

Response: We are grateful for Reviewer#1's helpful suggestions and have added more details to explain the parameters measured in Figure legend of Figure 5D as Reviewer#1 has suggested. Now it reads as follows: “(D) Measurement of serum indices including liver enzymes (AST, ALT and GGT) and kidney markers (CREA and UA) from *mdx* mice treated with PMO-M (n=4) or PMO alone (n=3) to reflect liver and kidney functions (*p<0.05).” Also the Figure has been renamed as Figure 6A and 6B.

Figure 5f: to be objective regarding the inflammatory response, quantification of CD68 should be done in the 4 groups. To be more complete other markers should be measured: eg CD4, CD8, CD11b.

Response: We thank Reviewer#1 for these comments and have quantified the number of CD68-positive macrophages in Figure 5F (renamed as 6D) and also provided as Figure 6E as reviewer#1 has suggested. In addition, to better reflect the inflammatory response, we have also measured CD3-positive T cells and CD11b-positive monocytes as Reviewer#1 has recommended and provided in Figure 6D and 6E.

Supplementary Figure 6c: Quantification of CD68 should be done as well to support the conclusion.

Response: We have quantified the number of CD68-positive macrophages in Supplementary Figure 6C and provided as Appendix Figure S4D as Reviewer#1 has suggested.

Point 6: Methods:

The method section could be improve. The authors should specify in the legend of the figures which cell lines were used: C2C12, H2K WT and mdx? Which H2K mdx cell lines was used (mutation exon 23, 52, else?), What is the PMO sequence used?

Response: We thank Reviewer#1 for these comments and have added more details in the corresponding Materials and Methods section as Reviewer#1 has suggested. Now it reads as follows: “PMO were synthesized and purified by GeneTools (Corvallis, OR, US). PMO (5'GGCCAAACCTCGGCTTACCTGAAAT 3') sequence was targeted to murine dystrophin exon 23/ intron 23 boundary sites as reported previously (Harding, Fall et al., 2007).

H2K mdx and WT cell lines: where these cell lines come from? Gift or bought? Which mutation H2K mdx carry? Which background of mice the cells have been generated on? Protocol for expansion and maintenance is incomplete.

Response: We have added more details in the corresponding Materials and Methods section. Now it reads as follows: “Immortalized murine H2K / and H2K-tsA58 *mdx* cells derived from *H-2K^b-tsA58* mice with a nonsense point mutation in exon23 of dystrophin were kindly provided by

Professor Terry Partridge (Children's National Medical Center, Center for Genetic Medicine Research, Washington DC, US) and cultured as previously reported (Morgan, Beauchamp et al., 1994). Briefly, cells were grown in Dulbecco's modified Eagle's medium (DMEM) supplemented with 20% fetal calf serum (FBS), 2% chick embryo extract, 2% L-glutamine, 1% penicillin and streptomycin and 20U/mL mouse recombinant IFN- γ (Invitrogen, US) at 33°C in 10% CO₂.”.

Why is there a paragraph on C2C12 cell line for PMO uptake while all the results are shown on dystrophic cells (so presumably on H2K-mdx cell lines)?

Response: We thank Reviewer#1 for pointing this out and have rectified the error in the corresponding Materials and Methods section.

Muscle endurance: for how long the mice were monitored in cage and when after the treatment?

Response: The treated and control mice were monitored for 48 hrs. For the treated mice, the test was conducted two weeks after last treatment. We have added more details in the corresponding Materials and Methods and Figure legends as Reviewer#1 has suggested.

Point 7: Page 4 : define GF

Response: We defined GF on Page 3, nevertheless we have re-defined GF on page 4 as Reviewer#1 has suggested.

Point 8: A space before the references in the text is often missing.

Response: We thank Reviewer#1 for these helpful suggestions and have carefully checked throughout the manuscript to avoid this.

Point 9: Figure 1A: same scale for the Y-axes should be used for H2K and H2K-mdx graphs to be able to compare them directly.

Response: We are grateful for Reviewer#1's helpful suggestion and have re-drawn the graph as Reviewer#1 has suggested.

Point 10: Page 5, the word anaerobic should be added in sentence as follow: "This result demonstrates that the increase in ECAR upon addition of glucose is due to anaerobic glycolysis rather than other sources."

Response: Again we are grateful for Reviewer#1's helpful suggestion and have added “anaerobic” in the sentence as Reviewer#1 has suggested.

Reviewer #2

Point 1: In the introduction section, in addition to eteplirsen, golodirsen is another PMO antisense oligonucleotide that has been approved by the FDA, and viltolarsen was approved in Japan. These need to be mentioned.

Response: We thank Reviewer#2 for the helpful suggestions and have changed the text accordingly. Now it reads as follows: “Although antisense oligonucleotide (AO)-mediated exon-skipping therapies show promise in DMD, with AO drugs - including eteplirsen, golodirsen and viltolarsen approved by the US FDA and in Japan (Frank, Schnell et al., 2020, Roshmi & Yokota, 2019, Syed, 2016) , limited systemic efficacy due to insufficient systemic delivery (Godfrey, Desviat et al., 2017) can be improved.”

Point 2: The authors used rhodamine B-labeled MOTS-c to quantify the levels in tissues but did not include TA muscles in Figure 2. It would be interesting to add TA here so that the correlation between the uptake and dystrophin levels can be examined.

Response: We are grateful for Reviewer#2's helpful comments and have added TA muscles in Figure 2B and 2C as Reviewer#2 has suggested. Based on the data, there was a marginal increase in the level of fluorescence intensity in TA muscles treated with PMO-M compared to PMO. Consistent with previous observations (Lin et al. *Molecular Therapy* (2020) 28(5):1339-1358; Han et al. *Nature Communications* (2016) 7:10981), the fluorescence intensity was relatively lower in TA muscles than quadriceps (Q), gastrocnemius (G), triceps (T) and abdominal muscles (A), which is likely due to the smaller volume of TA muscles. Quantification of PMO in TA muscles and gastrocnemius confirmed significantly increased uptake of PMO in peripheral muscles from PMO-M treated *mdx* mice compared to PMO-treated *mdx* mice (Figure EV1). We have provided the data as Figure EV1.

Also a correlation was established between the level of dystrophin expression in different muscles of *mdx* mice treated with PMO-M intravenously at the PMO dose of 50mg/kg/week for 3 weeks and tissues were examined two weeks after last injection (Figure 2H) and distribution of PMO-M (Figure 2C; 48hrs after single intravenous injection of FITC-labeled PMO (50mg/kg)) and / or MOTS-c (Figure 1D; 2hrs after single intravenous of Rhodamine B-labeled MOTS-c (500µg)) in different muscles of treated *mdx* mice.

Point 3: In the introduction section, the authors state that "repeated administration of MOTS c with extremely low doses of PMO (12.5mg/kg) elicited therapeutic levels of dystrophin restoration and functional improvements in *mdx* mice without any adverse effect ."; however, this is an overstatement as adverse effects were not examined extensively.

Response: We are grateful for Reviewer#2's comments and have re-phrased the sentence as Reviewer#2 has suggested. Now it reads as follows: "Moreover, repeated administration of MOTS-c with extremely low doses of PMO (12.5mg/kg) elicited therapeutic levels of dystrophin restoration and functional improvements in *mdx* mice without any detectable adverse effect, though more comprehensive toxicological studies are required prior to clinical deployment."

Point 4: the authors claim "uniform distribution of dystrophin-positive myofibers" (page 7) and "uniform expression of dystrophin" (page 8); however, these are confusing as the distribution appears to be mosaic rather than uniform in IHC.

Response: To make it clearer, the original statement on Page 7 was as follows: "Strikingly, a profound enhancement was observed in *mdx* mice treated with PMO-M as revealed by more uniform distribution of dystrophin-positive myofibres in peripheral muscles except for the heart (Figure S2A) and greater levels of dystrophin restoration (Figure S2B and S2C), compared to PMO alone under identical conditions." It means that more uniform dystrophin expression in muscles from PMO-M-treated *mdx* mice than PMO-treated *mdx* mice. Based on the results, there was more uniform distribution of dystrophin-positive fibres in peripheral muscles from *mdx* mice treated with PMO-M than PMO alone. Therefore, we believe the use of "uniform" on Page 7 is appropriate.

However, we have re-phrased the second "uniform" as Reviewer#2 has suggested. Now it reads as follows: "Surprisingly, widespread expression of dystrophin over multiple tissue sections

within each muscle group was detected in hind limb, fore limb, abdominal wall and diaphragm muscles, but not in the heart of *mdx* mice treated with repeated injections of PMO-M (Figure 4B).”

Point 5: The authors examined AST and ALT levels; however, GGT (gamma-GTP) is more appropriate to examine liver function as AST and ALT levels are already elevated in dystrophic models and can mask the toxic effects.

Response: We are grateful for Reviewer#2’s helpful suggestions and have examined the level of GGT as Reviewer#2 has recommended. And the data has been provided as Figure 6A.

Point 6: As authors employed peptides, immune response (e.g. CD3, CD4, CD8) should be examined.

Response: We are grateful for Reviewer#2’s comments and have examined CD3-positive T cells and CD11b-positive monocytes in treated and untreated muscle tissues as Reviewer#2 has recommended. And we have provided the data as Figure 6D and 6E.

Point 7: No western blotting included for the heart muscle. It can be incorporated to see if there is even a slight fold change in the dystrophin level in studying the long-term therapeutic efficacy because cardiomyopathy is predominantly prevalent in DMD patients.

Response: We thank Reviewer#2 for the helpful comments and have provided the Western blot results for the hearts as Reviewer#2 has recommended. The results showed trace amounts of dystrophin expression in the heart of PMO-M-treated *mdx* mice. We have provided the data as Appendix Figure S2.

Point 8: The authors need to mention the condition for the toxicity studies in figure 5 legend, as it is confusing to follow. (i.e. the timeline of the administration, concentration, etc.) Is it the long-term study at the lower dose of PMO?

Response: We thank Reviewer#2 for pointing this out and have added more details in the corresponding Figure legends for Figure 5 and Figure 6 as Reviewer#2 has suggested.

Point 9: Quantification and analysis of the myofibers (cross-section area, number of centrally nucleated fibers) are required in addition to the H and E staining of the diaphragm and quadriceps for the long-term study.

Response: We have provided the data of cross-sectional area and number of centrally-nucleated fibres for the long-term study as Figures 5D and 5E as Reviewer#2 has recommended.

Point 10: As the authors mentioned in the study that MOTS-c depletes intracellular 5-Methyltetrahydrofolate (5Me-THF) which is essential for cell division, a subsequent study might be necessary to determine the tolerated dose which doesn't lead to muscle wasting.

Response: We thank Reviewer#2 for this comment and have discussed this point in the Discussion. Now it reads as follows: “Nevertheless, a detailed study on the optimal titration of MOTS-c is warranted prior to its clinical deployment.”

Point 11: The sequence of the peptide and PMO need to be mentioned in the material-methods section.

Response: We are grateful for Reviewer#2’s comments and have added more details in the corresponding Materials and Methods as Reviewer#2 has recommended.

We very much hope that you will find our revised manuscript and detailed responses to the reviewers' comments satisfactory, and will now consider the manuscript suitable for publication in EMBO Molecular Medicine. We believe our manuscript to be of importance, principally as it shows that MOTS-c is an effective delivery vehicle for PMO and thus accelerate the clinical translation of PMO in DMD.

We look forward to hearing from you in due course.

With best wishes

HaiFang Yin

9th Nov 2020

Dear Prof. Yin,

Thank you for the submission of your revised manuscript to EMBO Molecular Medicine. I am pleased to inform you that we will be able to accept your manuscript pending the following final amendments:

Please implement all adjustments suggested by the referee #1.

***** Reviewer's comments *****

Referee #1 (Remarks for Author):

Thank you for the authors to try to answer my concerns - and most of them have be answered. The few comments that still need to be addressed are listed below.

I apology to not have been clear for Figure 2, point 3.

- Previous question: "Does the dystrophin expression level correlate with the PMO-M level observed in different muscle at 48h post-injection and/or with MOTS-c distribution? (the authors should have the info for the TA, Q, G, T, A, D, H)".

- Explanation: The authors already had all the data in the first version of the paper submitted. I was meaning to try to see if a correlation exist between dystrophin level shown in the western blot 2E and the level of PMO-M shown in figure 2C. I agree it is obviously not the same muscles being analysed for PMO-M quantity and dystrophin level, but as the results are tight (small error bars), it would be quite informative to understand the behaviour of PMO-M (is it just a relationship of quantity/efficacy or there is other parameters that may impact the efficiency of the PMO-M). Looking at the data, I suspect that the TA might be an outsider.

- The authors wrote that "Nevertheless, a correlation was established between the level of dystrophin expression in different muscles of mdx mice treated with PMO-M intravenously at the PMO...", but I can't see the actual correlation graph anywhere (best fit curve, R2, equation?).

Figure 3h (ex 3g) and 4d: error bar. Even if it is ratio, error bars can and should still be added. I understand the graph is "the fold change of PMO-M relative to PMO, in which the mean value of PMO-M and PMO was used". Individual PMO-M value can be divided by PMO mean, by doing this a SD can be generated.

Referee #2 (Comments on Novelty/Model System for Author):

The authors adequately addressed all the previous comments.

Referee #2 (Remarks for Author):

The authors adequately addressed all the previous comments.

The authors performed the requested editorial changes.

Referee #1

Thank you for the authors to try to answer my concerns - and most of them have been answered. The few comments that still need to be addressed are listed below.

Point 1: I apologize to not have been clear for Figure 2, point 3.

- Previous question: "Does the dystrophin expression level correlate with the PMO-M level observed in different muscle at 48h post-injection and/or with MOTS-c distribution? (the authors should have the info for the TA, Q, G, T, A, D, H)".

- Explanation: The authors already had all the data in the first version of the paper submitted. I was meaning to try to see if a correlation existed between dystrophin level shown in the western blot 2E and the level of PMO-M shown in figure 2C. I agree it is obviously not the same muscles being analysed for PMO-M quantity and dystrophin level, but as the results are tight (small error bars), it would be quite informative to understand the behaviour of PMO-M (is it just a relationship of quantity/efficacy or there are other parameters that may impact the efficiency of the PMO-M). Looking at the data, I suspect that the TA might be an outlier.

- The authors wrote that "Nevertheless, a correlation was established between the level of dystrophin expression in different muscles of mdx mice treated with PMO-M intravenously at the PMO...", but I can't see the actual correlation graph anywhere (best fit curve, R², equation?).

Response: We thank Referee #1 for the comments and have provided a correlation graph (Figure EV1E) as Referee #1 has suggested.

Point 2: Figure 3h (ex 3g) and 4d: error bar. Even if it is a ratio, error bars can and should still be added. I understand the graph is "the fold change of PMO-M relative to PMO, in which the mean value of PMO-M and PMO was used". Individual PMO-M value can be divided by PMO mean, by doing this a SD can be generated.

Response: We thank Referee #1 for the helpful suggestion and have modified Figures 3H and 4E as Referee #1 has recommended.

We very much hope that you will find our revised manuscript and detailed responses to the Referee#1's and editor's comments satisfactory, and will now consider the manuscript suitable for publication in EMBO Molecular Medicine.

We look forward to hearing from you in due course.

The authors performed the requested editorial changes.

Corresponding Author Name: HAIFANG YIN
Journal Submitted to: EMBO Molecular Medicine
Manuscript Number: EMM-2020-12993